

**Holocene glaciation in the Rwenzori Mountains, Uganda**
Margaret S. Jackson[1]*, Meredith A. Kelly[1], James M. Russell[2], Alice M. Doughty[3], Jennifer A.
Howley[1]^,  Susan R.H. Zimmerman[4], Bob Nakileza[5]
1. Earth Sciences, Dartmouth College, Hanover, NH, United States
2. Earth, Environmental, and Planetary Sciences, Brown University, Providence, RI United
States
3. Geology, Bates College, Lewiston, ME
4. Center for Accelerator Mass Spectrometry, Lawrence Livermore National Laboratory,
Livermore, CA United States
5. Mountain Resource Centre, Makerere University, Kampala, Uganda
*Current address: School of Geography, Archaeology, and Irish Studies, National University of
Ireland Galway, Galway, Ireland
^Current address: New Hampshire Department of Health and Human Services, Concord, NH,
United States
Correspondence to: Margaret S. Jackson (margaret.jackson@nuigalway.ie)





**Abstract**

Tropical glaciers are retreating rapidly, threatening alpine ecosystems across the low latitudes.

Understanding how tropical glaciers responded to past periods of warming is crucial for

predicting and adapting to future climate change, yet relatively little is known about glacial

fluctuations in tropical regions during the recent past (i.e., the Holocene Epoch). This is

particularly true in the African tropics, where data constraining the timing and magnitude of

Holocene glacial fluctuations in the region are sparse and where temperatures during the Middle

Holocene were perhaps as warm as or warmer than today. Here we present new beryllium-10

surface-exposure ages that constrain Holocene glacial extents in the equatorial Rwenzori

Mountains, Uganda. These results document rapid Early Holocene (~11.7-8.2 ka) glacial retreat

in two separate catchments and indicate that Late Holocene (~4.2 ka-present) deposits mark the

greatest expansion of Rwenzori glaciers during the last ~11 ka. Holocene glacial fluctuations

elsewhere in tropical Africa and in tropical South America are broadly similar to those in the

Rwenzori, with most tropical glaciers retreating rapidly during the Early Holocene and

remaining near or inboard of their Late Holocene positions through much of Holocene time. The

similarity of Holocene glacial fluctuations across the tropics implies that low-latitude glaciers

responded to a common forcing mechanism, most likely temperature. Although the drivers of

Holocene temperature changes in the tropics remain enigmatic, these data help constrain the

expression of tropical temperature changes in the low latitudes.



## 1 Introduction


The ongoing, coherent retreat of Earth's alpine glaciers is unique within the Holocene
Epoch (~11.7 ka-present; Walker et al., 2012) and emblematic of anthropogenic warming
(Solomina et al., 2015). The loss of alpine glaciers is of particular concern in the tropics, where
high-elevation regions are warming at a rate twice the global average (Vuille et al., 2008).
Tropical glaciers are a primary source of freshwater and are a fundamental component of
regional economies, underpinning agriculture, hydropower, and tourism (Bradley et al., 2006;
Chevallier et al., 2011). Accurately projecting the response of glaciers to future climate change is
thus crucial for effective community response and adaptation (Stocker et al., 2013), and these
projections rely on robust understanding of the sensitivity of tropical glaciers to past climate
conditions.
Tropical glaciers respond to changes in both temperature and precipitation, although the
relative influence of these forcings depends upon a glacier's unique climatic setting (Sagredo et
al., 2014). Recent work to reconstruct past glacial fluctuations in tropical South America
indicates that glaciers there were near or inboard of their Late Holocene (~4.2 ka-present; Walker
et al., 2012) maxima during much of the Holocene Epoch (Jomelli et al., 2014; Solomina et al.,
2015; Stansell et al., 2017). Although relatively little is known about Holocene glacial
fluctuations in the African tropics (Kaser and Osmaston, 2002; Solomina et al., 2015), recently
produced terrestrial paleotemperature reconstructions provide greater paleoclimatic context for
understanding past changes in tropical African glacial extent (Weijers et al., 2007; Tierney et al.,
2008; Woltering et al., 2011; Loomis et al., 2012, 2017). Of particular interest is the response of
glaciers to climate conditions during the Middle (~8.2-4.2 ka) and Early (~11.7-8.2 ka) Holocene
Epoch (Walker et al,. 2012), when temperatures in tropical Africa may have been similar to or





higher than modern (Ivory et al., 2017 and references therein). Determining when and how
glaciers in the African tropics fluctuated during past warm periods provides crucial information
for assessing whether, or how long, tropical glaciers may persist under future warming scenarios.

Here we present new data from the equatorial Rwenzori Mountains of Uganda (0.3ºN,

30.0ºE; Figure 1) that constrain the extent of glaciers in two separate valleys during the
Holocene. These data include twelve beryllium-10 ($^{10}$Be) surface-exposure ages of glacial
landforms which provide evidence of past glacial extents in the Rwenzori during Holocene time.
We then compare the Rwenzori glacial chronology with records of East African glaciation and
paleoclimate to assess both the potential drivers of past glacial fluctuations as well as the
response of these glaciers to changes in Holocene climate.

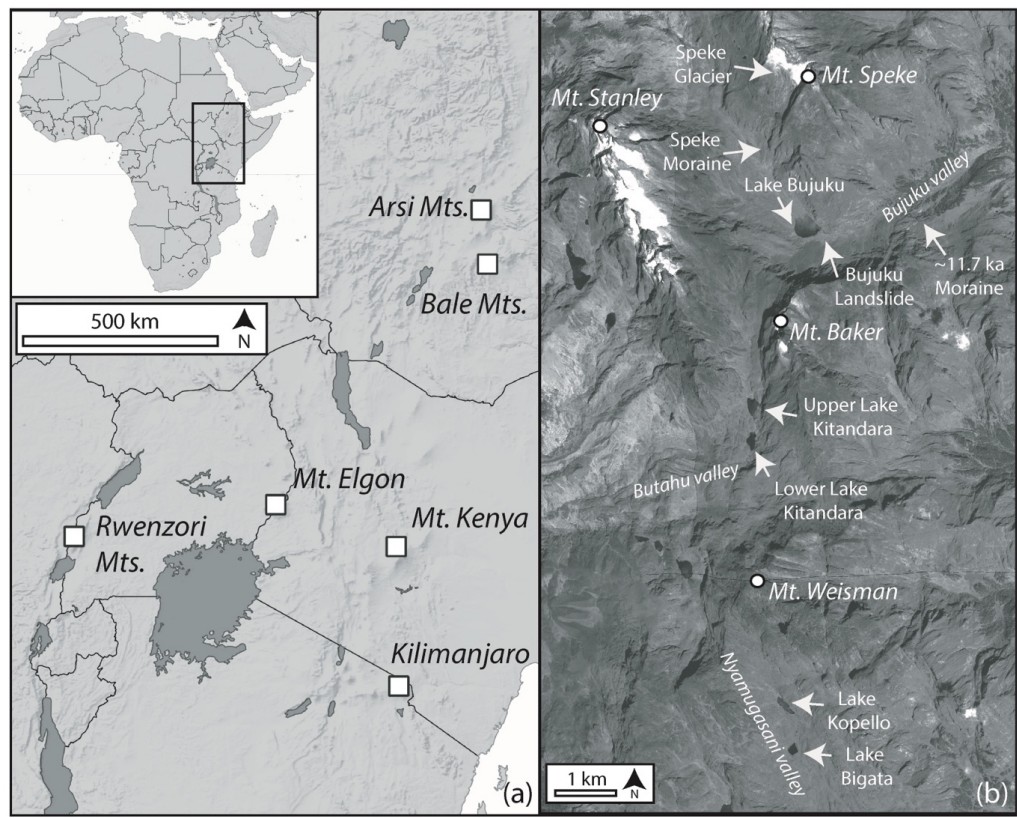

**Figure 1.** Tropical East Africa and the Rwenzori Mountains. (a) The Rwenzori Mountains, Kilimanjaro, and Mt. Kenya are the only three still-glacierized sites in East Africa. Mt. Elgon in Uganda and the Arsi and Bale Mountains in Ethiopia also host glacial deposits, though are no longer glacierized. (b) Worldview-1 satellite image of the central Rwenzori massif and locations mentioned in the text. Glaciers persist in the Rwenzori on Mt. Stanley, Mt. Speke, and Mt. Baker. Although no longer glacierized, the former Thomson Glacier occupied the peak of Mt. Weisman during the early 20th century.

## 2 Background

The Rwenzori Mountains are an uplifted horst of crystalline bedrock and are an extreme example of rift shoulder uplift (McConnell et al., 1953; Ring, 2008). Mt. Stanley, the highest point in the range, reaches an elevation of 5,109 meters above sea level (m asl) and, with Mt. Speke and Mt. Baker, is one of three still-glacierized peaks in the Rwenzori (Figure 1). The first quantitative observations of Rwenzori glaciers were made in the early 20th century (Abruzzi,



1907). Rwenzori glaciers have retreated markedly since then, decreasing in area from ~6.5 to
0.96 km² between 1906 and 2003 CE and losing an estimated 50% of their areal extent between
1987 and 2003 CE (Taylor et al., 2006). Modern glaciers in the Rwenzori occur only above
~4,400 m asl and are predicted to disappear within the coming decades (Kaser and Osmaston,
2002; Taylor et al., 2006).

Although at present glaciers occupy only the highest peaks (Figure 1), the Rwenzori host

glacial deposits that attest to more extensive glaciation during and since the last ice age.
Osmaston (1965; 1989) grouped moraines in the Rwenzori into distinct glacial stages based on
their relative weathering, stratigraphic position, and morphology. The 'Omurubaho' stage
moraines occur at elevations ~3,600-4,000 m asl and feature ~3-30 m relief above the valley
floors (Osmaston, 1989). Osmaston (1989) estimated these moraines to have been deposited
during the Early Holocene. Recent [10]Be dating of Omurubaho stage moraines in the Bujuku and
Nyamugasani valleys indicates deposition during late-glacial (~15.0-11.7 ka) and Early
Holocene time (Jackson et al., in review). The 'Lac Gris' stage moraines (Osmaston, 1989) are
located up valley and stratigraphically inboard of the Omurubaho stage moraines. Lac Gris stage
moraines are predominantly low-relief features (1-2 m above the valley floors) and are within
~100 m of observed 1906 CE ice extents (Abruzzi, 1907). Osmaston (1989) estimated Lac Gris
stage moraines to be ~700-100 years old. Based on lichenometry, Bergström (1955) suggested
that Lac Gris stage moraines observed near the margin of Elena Glacier on Mt. Stanley date to
~1750 CE. However, the rate at which lichens colonize rock surfaces in the Rwenzori is
unconstrained (Osmaston et al., 1989) and the ultimate age of pre-observation Lac Gris moraines
remains undetermined.





Livingstone (1967) obtained radiocarbon ages of lake sediments that provide minimum
limits on the timing of deglaciation at several locations in the Rwenzori. In the Butahu valley,
dated organic sediments within a horizon roughly one meter above presumed basal silts in Upper
Lake Kitandara (4,000 m asl; Figure 1) yield a radiocarbon age of ~7.7 cal kyr BP (Livingstone,
1967). This provides a minimum-limiting age for glacial recession past this location in the
valley. In the Bujuku valley, one radiocarbon age from a layer of gravel-rich peat in a sediment
core from Lake Bujuku (3,920 m asl) yields an age of ~3.1 cal kyr BP (Livingstone et al, 1967).
However, this core did not recover the complete sedimentary succession from the lake and,
therefore, may significantly postdate the onset of post-glacial sedimentation in the lake.
Rwenzori glacial fluctuations that occurred between the Early Holocene and the (near) historical
period are largely unconstrained. A sediment core from Lower Lake Kitandara (~4,000 m asl)
indicates changes in lake water chemistry at the turn of the 18th century consistent with greater
glacier meltwater flux to the lake, although with no corresponding changes in pollen or diatom
assemblages (McGlynn et al., 2010). High-resolution analyses of clastic sediment input to
numerous Rwenzori alpine lakes indicate that recent historical glacial recession began about
1870 CE (Russell et al., 2009).
Evidence from elsewhere in tropical East Africa suggests that glaciers across the region
were near or inboard of their maximum Late Holocene extents for much of Holocene time.
Radiocarbon ages from lake sediments in the Bale Mountains and on Mt. Arsi in the Ethiopian
Highlands, as well as from Mt. Elgon in Uganda, indicate that glaciers at these sites were inboard
of their late-glacial extents, and perhaps had ablated completely, by the Early Holocene
(Hamilton and Perrot, 1982; Tiercelin et al., 2008). In Tanzania, the persistence of Holocene ice
cover on Kilimanjaro is a subject of ongoing debate. Multiple studies show that the Kilimanjaro





Ice Cap has become less extensive over the last ~1 ka, although whether the ice cap may have
ablated completely at some point during the Holocene and later re-nucleated is uncertain
(Thompson et al., 2002; Kaser et al., 2010; Thompson et al., 2011; Noell et al., 2014). Recent
radiocarbon dating of dust and soil horizons within the ice cap suggests a period of net ice cap
ablation occurred prior to ~4 ka, followed by net accumulation during the Late Holocene that
persisted until near-historical time when recent recession began (Gabrielli et al., 2014).

At Mt. Kenya, chlorine-36 ($^{36}$Cl) surface-exposure dating of moraines and glacially

molded bedrock in the Teleki Valley indicates that glaciers retreated from their Early Holocene
maximum extents by ~10 ka and that the modern Lewis Glacier reached its maximum Late
Holocene extent ~200 years ago (Shanahan and Zreda, 2000). In contrast to the existing evidence
for limited Middle Holocene glacial expansion in the Rwenzori and on Kilimanjaro, radiocarbon
ages of Mt. Kenya's Naro Moru Tarn moraine dam, located ~250 m down valley of the ~200
year old Lewis Glacier moraine, suggest a glacial advance occurred in the Teleki Valley between
~6.9 and 4.7 cal yr BP (Johansson and Holmgren, 1985; Karlen et al., 1999). Although disputed
(Mahaney et al., 1989), radiocarbon ages from Thomson Tarn in the Hobley Valley also suggest
a Middle Holocene glacial advance on Mt. Kenya between ~7.1 and 6.2 cal yr BP (Perrot, 1982).
In addition, clastic sediment fluxes to high alpine lakes on Mt. Kenya indicate more dynamic,
erosive glacial activity after ~5 cal yr BP (Karlen et al., 1999), although the extent of
corresponding glacial margins throughout this period is not known.

This study aims to establish the timing of Holocene glacial fluctuations in the Rwenzori

Mountains and compare directly the Holocene Rwenzori glacial chronology with records of
glaciation and paleoclimate elsewhere in tropical East Africa. Prior work mapping and dating
Last Glacial Maximum (LGM) and late-glacial Rwenzori glacial deposits lends crucial temporal



and spatial context for the Holocene chronology (Kelly et al., 2014; Jackson et al., 2019; Jackson
et al., in review) and allows a more robust comparison of the Rwenzori glacial record with
records of past regional climate conditions.

## 3 Study Sites

We focused our study within the Bujuku and Nyamugasani valleys, two independent

catchments in the Rwenzori that contain Holocene-age glacial deposits amenable for [10]Be dating
and for which there is pre-existing numerical age control on pre- or Early Holocene glacial
deposits (Jackson et al., 2019; Jackson et al., in review).

### 3.1 Bujuku valley

The modern Speke Glacier occupies the south-facing peak of Mt. Speke (4,890 m asl)

near the head of the Bujuku valley (Figure 1, 2). Although today the glacial terminus occurs at
~4,600 m asl, in 1958 CE the glacier extended down slope to ~4,350 m asl (Whittow et al.,
1963). Lac Gris stage deposits occur on Mt. Speke between ~4,000 and 4,500 m asl (Osmaston,
1989), including a Lac Gris stage moraine ~300 m downslope from the 1958 CE glacial extent
(Osmaston, 1989)(Figure 2). Observations by Abruzzi (1907) indicate that ice had abandoned
this Lac Gris stage moraine prior to 1906 CE, though the precise timing of retreat is not known
(Whittow, 1963).

Omurubaho stage moraines occur ~2.5 km down the Bujuku valley from the Lac Gris

stage moraines, and the innermost (i.e., farthest up valley) of these dates to ~11.7 ka (Jackson et
al., in review). There are no moraines in the valley between the ~11.7 ka moraine and the Lac
Gris stage deposits on Mt. Speke. Approximately 1.5 km up valley from the ~11.7 ka moraine,



the outlet of Lake Bujuku is dammed by a landslide that originated on the north-facing slope of
Mt. Baker (Figure 1). This landslide is dated to ~11 ka and shows no evidence of having been
impeded or reworked by ice either during or subsequent to deposition (Cavagnaro, 2017).
Therefore, it is likely that the landslide was emplaced after ice retreated up valley and the age
(~11 ka) is a minimum-limiting age for deglaciation of the valley floor at this location (Jackson
et al., in review).

**3.2 Nyamugasani valley**

194        Mt. Weisman (4,620 m asl) marks the head of the Nyamugasani valley (Figure 1, 2).

Although no longer glacierized, the former Thomson Glacier occupied a cirque on the south-
facing slope of the peak until the mid-20th century (Osmaston and Pasteur, 1972). Omurubaho-
stage moraines occur in the Nyamugasani valley between ~3,800 and 4,000 m asl. The innermost
(i.e., farthest up valley) Omurubaho moraine is dated to ~11.2 ka and dams Lake Bigata (~4,000
m asl)(Jackson et al., in review). There are no moraines between Lake Bigata and the peak of Mt.
Weisman, although glacially transported boulders are ubiquitous on the valley
floor. Approximately 0.5 km up valley from the ~11.2 ka moraine, four boulders on a bedrock
rise at the outlet of Lake Kopello (~4,020 m asl) yield ages between ~12.1 and 10.5 ka,
indicating continued recession of ice in the valley after ~11 ka (Jackson et al., in review).

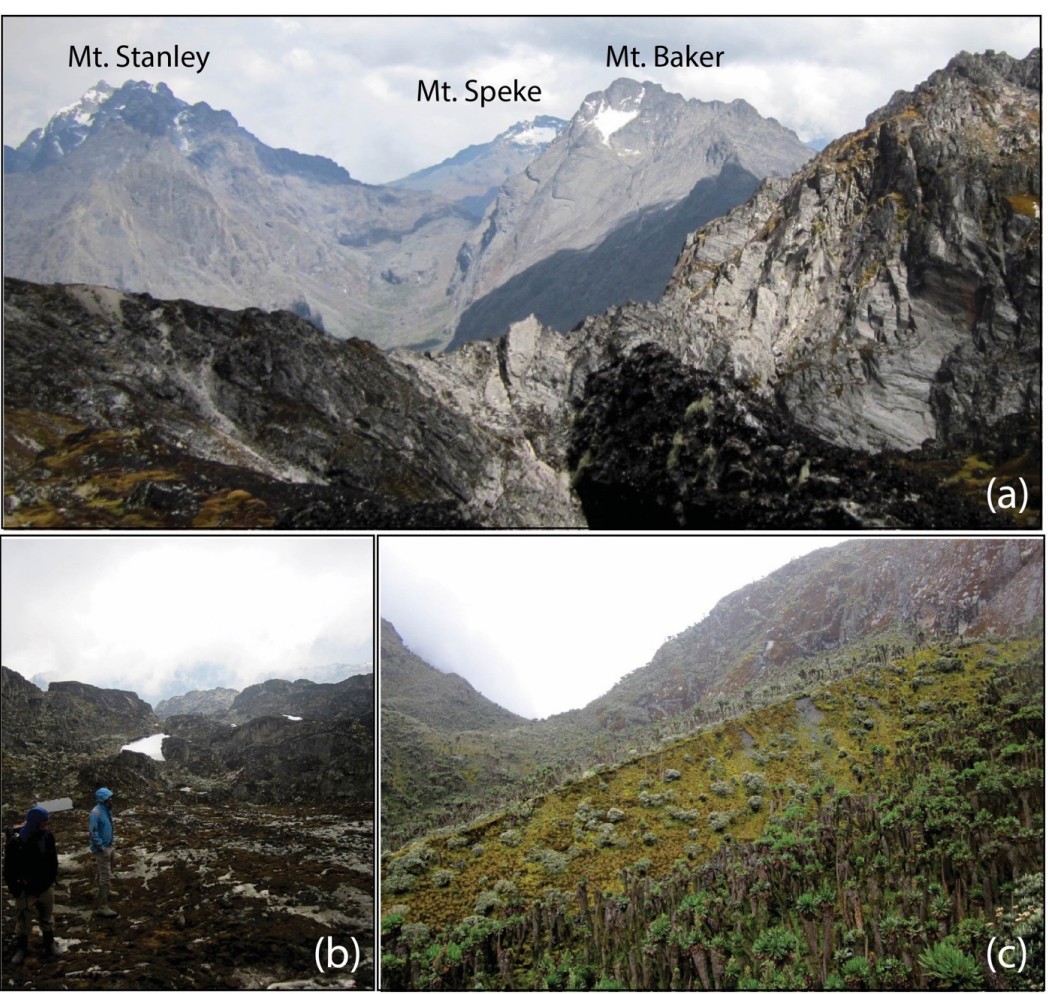


**Figure 2.** Glacial features in the central Rwenzori Mountains. (a) A view toward the north from the peak of Mt. Weisman. Mt. Stanley, Mt. Speke, and Mt. Baker are the three still-glacierized peaks in the Rwenzori. Above 4,000 m asl, the Rwenzori are dominated by bare bedrock with some lichen and moss cover. (b) A view down valley, toward the south, from the unoccupied cirque on Mt. Weisman that once held the former Thomson Glacier. (c) The right-lateral Speke moraine beneath Speke Glacier features a sharp crest and steep ice-proximal slope. View is toward the east.





N/A




**4 Methods**
We conducted three field seasons in the Rwenzori between 2012 and 2016. While in the
field we classified glacial-geomorphic features based on their morphology, stratigraphic position,
and degree of weathering and mapped these features onto WorldView-1 0.5-m resolution
satellite imagery. We collected samples for [10]Be dating from boulders on moraines, boulders on
bedrock, and from bedrock surfaces using a hammer and chisel and the drill-and-blast method of
Kelly (2003). We took care to sample boulders that showed no indication of post-depositional
movement and, where possible, surfaces with no dip in order to minimize topographic shielding
correction uncertainties. We recorded sample locations with a handheld GPS (± 3 m vertical, ± 1
m horizontal), determined topographic shielding using a clinometer, and measured sample
surface dip and dip direction, if applicable, with a handheld compass (Table 1).

**Table 1: Rwenzori Sample Information**

**Bujuku Valley**

| Map ID | Sample ID | Landform | Latitude (DD) | Longitude (DD) | Elev. (m) | Atm. | Thickness (cm) | Density (g/cm3) | Shielding | Erosion (mm/yr) | 10-Be (atoms/g) | ± 10-Be (atoms/g) |
|---|---|---|---|---|---|---|---|---|---|---|---|---|
| 1 | RZ-12-21 | Speke moraine | 0.38750 | 29.88821 | 4095 | std | 1.5 | 2.65 | 0.909 | 0 | 9.29E+03 | 7.84E+02 |
| 2 | RZ-12-22 | Speke moraine | 0.38768 | 29.88816 | 4046 | std | 1.2 | 2.65 | 0.909 | 0 | 6.96E+03 | 4.80E+02 |
| 3 | RZ-12-24 | Speke moraine | 0.38768 | 29.88816 | 4046 | std | 1.3 | 2.65 | 0.909 | 0 | 1.02E+04 | 5.07E+02 |
| 4 | RZ-12-25 | Speke moraine | 0.38768 | 29.88816 | 4046 | std | 1.5 | 2.65 | 0.909 | 0 | 1.17E+04 | 3.87E+02 |

**Nyamugasani Valley**

| Map ID | Sample ID | Landform | Latitude (DD) | Longitude (DD) | Elev. (m) | Atm. | Thickness (cm) | Density (g/cm3) | Shielding | Erosion (mm/yr) | 10-Be (atoms/g) | ± 10-Be (atoms/g) |
|---|---|---|---|---|---|---|---|---|---|---|---|---|
| 5 | RZ-15-10 | Perched boulder | 0.32265 | 29.89128 | 4397 | std | 4.0 | 2.65 | 0.976 | 0 | 3.78E+05 | 3.56E+03 |
| 6 | RZ-15-11 | Perched boulder | 0.32263 | 29.89132 | 4400 | std | 2.0 | 2.65 | 0.976 | 0 | 3.60E+05 | 2.50E+03 |
| 7 | RZ-15-09 | Perched boulder | 0.32385 | 29.89034 | 4431 | std | 3.0 | 2.65 | 0.983 | 0 | 3.51E+05 | 3.79E+03 |
| 8 | RZ-15-07 | Perched boulder | 0.32589 | 29.88928 | 4488 | std | 1.9 | 2.65 | 0.989 | 0 | 1.51E+05 | 1.50E+03 |
| 9 | RZ-15-08 | Perched boulder | 0.32601 | 29.88953 | 4498 | std | 2.0 | 2.65 | 0.99 | 0 | 2.19E+05 | 4.14E+03 |
| 10 | RZ-15-01 | Cirque Bedrock | 0.32793 | 29.88877 | 4509 | std | 1.9 | 2.65 | 0.969 | 0 | 1.66E+05 | 1.81E+03 |
| 11 | RZ-15-02 | Cirque Bedrock | 0.32786 | 29.88887 | 4526 | std | 1.4 | 2.65 | 0.97 | 0 | 1.69E+05 | 1.84E+03 |
| 12 | RZ-15-03 | Cirque Bedrock | 0.32781 | 29.88871 | 4536 | std | 2.8 | 2.65 | 0.97 | 0 | 1.89E+05 | 1.68E+03 |

**Table 1.** Geographic data and sample characteristic information for Rwenzori samples.





We isolated beryllium from each sample and associated process blanks at the Dartmouth

College Cosmogenic Nuclide Laboratory using a modified version of the methods described in

Schaefer et al. (2009). All $^{10}Be/^9Be$ ratios were measured at the Lawrence Livermore Center for

Accelerator Mass Spectrometry and normalized to the 07KNSTD3110 standard (Nishiizumi et

al., 2007)(Table 2). $^{10}Be$ ages presented in Figures 3-5 and in Table 3 are as calculated using

version 3 of the online calculator described by Balco et al. (2008 and subsequently updated) with

a high-altitude, low-latitude production rate (Kelly et al., 2015) and time-invariant scaling

framework ("St" scaling; Lal, 1991; Stone, 2000). We present $^{10}Be$ ages calculated using an

alternative, time-variant scaling framework ("LSDn" scaling; Lifton et al., 2016) in Table 3. Our

choice of scaling framework does not alter our overall interpretations. Where $^{10}Be$ concentrations

are of bedrock rather than glacially deposited sediments, we report the nuclide concentration

rather than the exposure-age equivalent (Figure 4, Table 1), as bedrock nuclide concentrations

may reflect multiple periods of exposure rather than a single exposure duration.


**Table 2: Rwenzori Sample Chemistry**

**Bujuku Valley**

| Map ID | Sample ID | Landform | Cathode ID | Quartz (g) | Carrier wt. (mg) | Carrier Conc. (ppm) | Sample (10-Be/9-Be) | ± Sample (10-Be/9-Be) | Process Blank Cathode ID | Blank (10-Be/9-Be) | ± Blank (10-Be/9-Be) |
|---|---|---|---|---|---|---|---|---|---|---|---|
| 1 | RZ-12-21 | Speke moraine | BE43754 | 16.1727 | 0.2013 | 0.973 | 1.14775E-14 | 9.69152E-16 | BE43758 | 1.63761E-15 | 3.53685E-16 |
| 2 | RZ-12-22 | Speke moraine | BE43755 | 32.3243 | 0.2006 | 0.973 | 1.72402E-14 | 1.19007E-15 | BE43758 | 1.63761E-15 | 3.53685E-16 |
| 3 | RZ-12-24 | Speke moraine | BE43756 | 23.8116 | 0.2013 | 0.973 | 1.85318E-14 | 9.2237E-16 | BE43758 | 1.63761E-15 | 3.53685E-16 |
| 4 | RZ-12-25 | Speke moraine | BE43757 | 40.0454 | 0.2010 | 0.973 | 3.59382E-14 | 1.18435E-15 | BE43758 | 1.63761E-15 | 3.53685E-16 |

**Nyamugasani Valley**

| Map ID | Sample ID | Landform | Cathode ID | Quartz (g) | Carrier wt. (mg) | Carrier Conc. (ppm) | Sample (10-Be/9-Be) | ± Sample (10-Be/9-Be) | Process Blank Cathode ID | Blank (10-Be/9-Be) | ± Blank (10-Be/9-Be) |
|---|---|---|---|---|---|---|---|---|---|---|---|
| 5 | RZ-15-10 | Perched boulder | BE39810 | 100.945 | 0.0907 | 1.338 | 4.70348E-12 | 4.43266E-14 | BE39812 | 3.81273E-15 | 6.15885E-16 |
| 6 | RZ-15-11 | Perched boulder | BE39811 | 102.028 | 0.091 | 1.338 | 4.51587E-12 | 3.13429E-14 | BE39812 | 3.81273E-15 | 6.15885E-16 |
| 7 | RZ-15-09 | Perched boulder | BE39809 | 100.573 | 0.0916 | 1.338 | 4.30701E-12 | 4.65808E-14 | BE39812 | 3.81273E-15 | 6.15885E-16 |
| 8 | RZ-15-07 | Perched boulder | BE39808 | 101.292 | 0.0881 | 1.338 | 1.94507E-12 | 1.93123E-14 | BE39812 | 3.81273E-15 | 6.15885E-16 |
| 9 | RZ-15-08 | Perched boulder | BE40319 | 12.014 | 0.1650 | 1.340 | 1.78244E-13 | 3.36358E-15 | BE40308 | 7.21905E-16 | 1.41075E-16 |
| 10 | RZ-15-01 | Cirque Bedrock | BE39531 | 100.57 | 0.0961 | 1.337 | 1.94043E-12 | 2.11829E-14 | BE39534 | 6.99989E-15 | 5.814E-16 |
| 11 | RZ-15-02 | Cirque Bedrock | BE39532 | 100.79 | 0.0967 | 1.337 | 1.97014E-12 | 2.14812E-14 | BE39534 | 6.99989E-15 | 5.814E-16 |
| 12 | RZ-15-03 | Cirque Bedrock | BE39533 | 101.33 | 0.0930 | 1.337 | 2.30326E-12 | 2.04419E-14 | BE39534 | 6.99989E-15 | 5.814E-16 |

**Table 2.** Processing data and sample chemistry for all Bujuku and Nyamugasani valley samples.





We do not correct [10]Be ages for the potential impacts of snow cover or vegetation. Snow
does not persist for considerable lengths of time at the sample elevations due to warm daytime
temperatures and intense solar radiation. Vegetation in the Rwenzori Mountains above ~4,000 m
asl is sparse (Osmaston and Pasteur, 1972; Foster et al., 2001) and all samples were collected
above this elevation. Some samples featured a patchy cover of lichen or moss (≤ 2 cm thick) and
we avoided this where possible, although we note that a persistent cover of moss of 2 cm
thickness would alter the resultant exposure ages by < 2% (Dunai, 2010; Plug et al., 2007). We
also did not correct [10]Be ages for the potential influence of erosion, as samples did not show
evidence that could be used to estimate quantitatively surface erosion rates. Previous applications
of [10]Be dating in the Rwenzori suggest that raised quartz veins and rock surfaces on single
moraine crests yield statistically similar ages (Jackson et al., 2019).






| Table 3: Rwenzori Surface-Exposure Ages | | | | | | | | |
|---|---|---|---|---|---|---|---|---|
| **Bujuku Valley** | | | | | | | | |
| Map ID | Sample ID | Landform | Age (St) | ± (int; St) | ± (ext; St) | Age (LSDn) | ± (int; LSDn) | ± (ext; LSDn) |
| 1 | RZ-12-21 | Speke moraine | 360 | 30 | 40 | 370 | 30 | 40 |
| 2 | RZ-12-22 | Speke moraine | 270 | 20 | 20 | 280 | 20 | 30 |
| 3 | RZ-12-24 | Speke moraine | 400 | 20 | 30 | 410 | 20 | 30 |
| 4 | RZ-12-25 | Speke moraine | 460 | 20 | 30 | 480 | 20 | 30 |
| | | | | | | | | |
| **Nyamugasani Valley** | | | | | | | | |
| Map ID | Sample ID | Landform | Age (St) | ± (int; St) | ± (ext; St) | Age (LSDn) | ± (int; LSDn) | ± (ext; LSDn) |
| 5 | RZ-15-10 | Perched boulder | 12130 | 120 | 690 | 11510 | 110 | 680 |
| 6 | RZ-15-11 | Perched boulder | 11360 | 80 | 650 | 11010 | 80 | 640 |
| 7 | RZ-15-09 | Perched boulder | 10920 | 120 | 630 | 10700 | 120 | 630 |
| 8 | RZ-15-07 | Perched boulder | 4520 | 50 | 260 | 4980 | 50 | 290 |
| 9 | RZ-15-08 | Perched boulder | 6520 | 120 | 390 | 6590 | 130 | 400 |
| | | | | | | | | |
| 10 | RZ-15-01 | Cirque Bedrock | 5010 | 60 | 290 | 5400 | 60 | 320 |
| 11 | RZ-15-02 | Cirque Bedrock | 5040 | 60 | 290 | 5430 | 60 | 320 |
| 12 | RZ-15-03 | Cirque Bedrock | 5680 | 50 | 320 | 5930 | 50 | 350 |

**Table 3.** Cosmogenic [10]Be surface-exposure ages for samples from the Bujuku and Nyamugasani valleys. We report ages as calculated using both time-invariant ("St"; Lal, 1991; Stone, 2000) and time-variant ("LSDn"; Lifton et al., 2016) scaling with internal ("int") and external ("ext") error.

## 5 Results

### 5.1 Bujuku valley

We term the prominent Lac Gris stage moraine on the south-facing slope of Mt. Speke the 'Speke moraine' (Figures 2-3). The Speke moraine marks the first glacial deposit up valley of the Omurubaho-stage moraine dated to ~11.7 ka (Jackson et al., in review) and is ~300 m elevation downslope from the 1958 CE glacier extent (Whittow, 1963). The Speke moraine is also ~1.5 km up valley of the ~11.0 ka landslide that dams Lake Bujuku (Cavagnaro, 2017). The right- and left-lateral ridges of the Speke moraine are well preserved and have steep ice-contact slopes with more fan-like, low-angle ice-distal slopes. Four samples from the right-lateral ridge yield [10]Be ages between ~0.46 and 0.27 ka (RZ-12-21, 22, 24, 25; ~4,050 m asl)(Table 1).



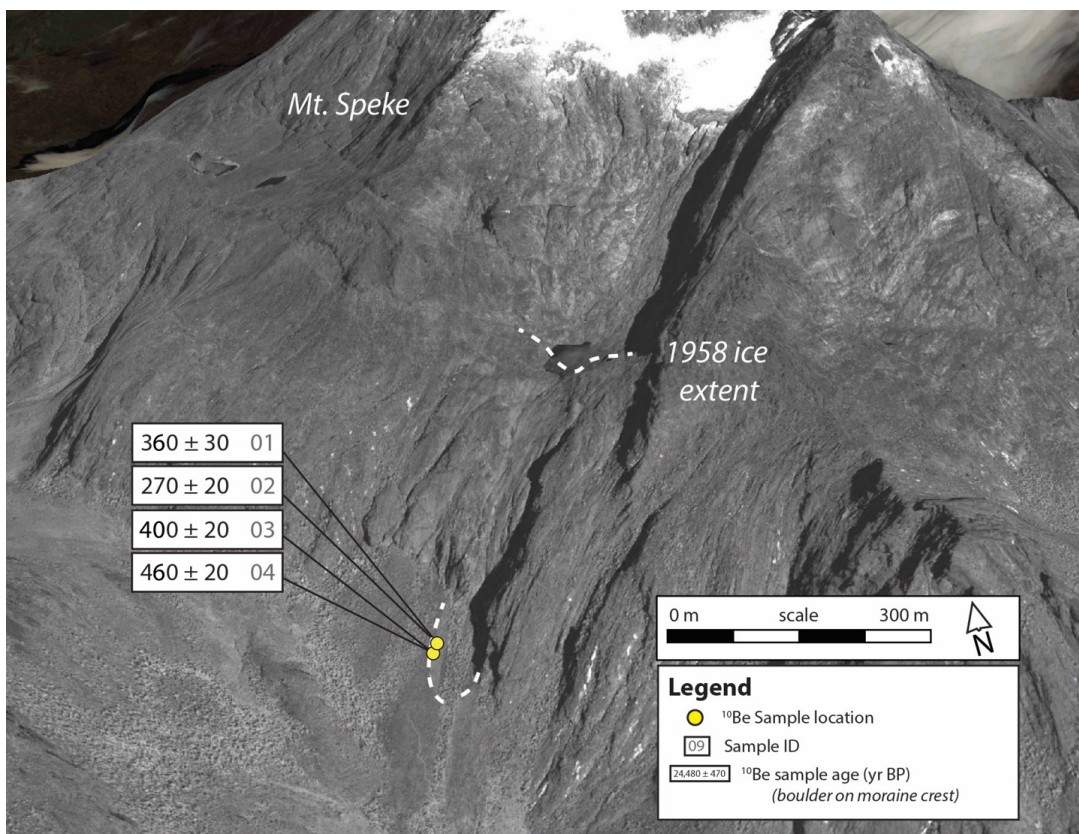

**Figure 3.** Mount Speke and [10]Be ages of the Speke moraine. Ages are shown in black with internal analytical
uncertainties. Sample ID numbers as in Tables 1-3 are shown in grey. The ice-contact slope of the Speke moraine is
outlined in dashed white, as is the documented 1958 ice margin ~ 300 m upslope (Whittow et al., 1963). Sample
locations are mapped onto a 0.5 m-resolution Worldview-1 satellite image.


**5.2 Nyamugasani valley**

We dated five boulders on bedrock along an elevation transect on the south-facing slope

of Mt. Weisman (Figure 4, Tables 1-3). The transect extends from ~4,400 to 4,490 m asl. All
samples are from glacially molded, sub-rounded boulders except for sample RZ-15-08 which is
from an angular boulder. Based on their size and position on the slope (i.e., away from the valley
walls where rockfall may occur), we presume these boulders were deposited by a receding
glacier. The two most down valley samples yield [10]Be ages of 12.1 ± 0.1 ka (RZ-15-10; 4,397m





asl) and 11.4 ± 0.1 ka (RZ-15-11; 4,400 m asl). A third sample is located ~170 m up valley at
~4,430 m asl and dates to 10.9 ± 0.1 ka (RZ-15-09). Approximately 250 m farther up valley at
~4,490 m asl, two boulders on bedrock knobs yield ages of 4.5 ± 0.0 ka (RZ-15-07) and 6.5 ± 0.1
ka (RZ-15-08).

In addition to the elevation transect of boulders on bedrock, we measured $^{10}$Be

concentrations in three samples of the bedrock floor of the unoccupied cirque below the peak of
Mt. Weisman (4,509-4,536 m asl)(Figures 2, 4). We term this feature the 'Thomson cirque'. The
bedrock samples from Thomson cirque contain $^{10}$Be concentrations between 1.94 and 2.03 x $10^{-12}$
atoms/gram (quartz), equivalent to ~5.0-5.7 thousand years of exposure (RZ-15-01, 02,
03)(Table 1, 3).





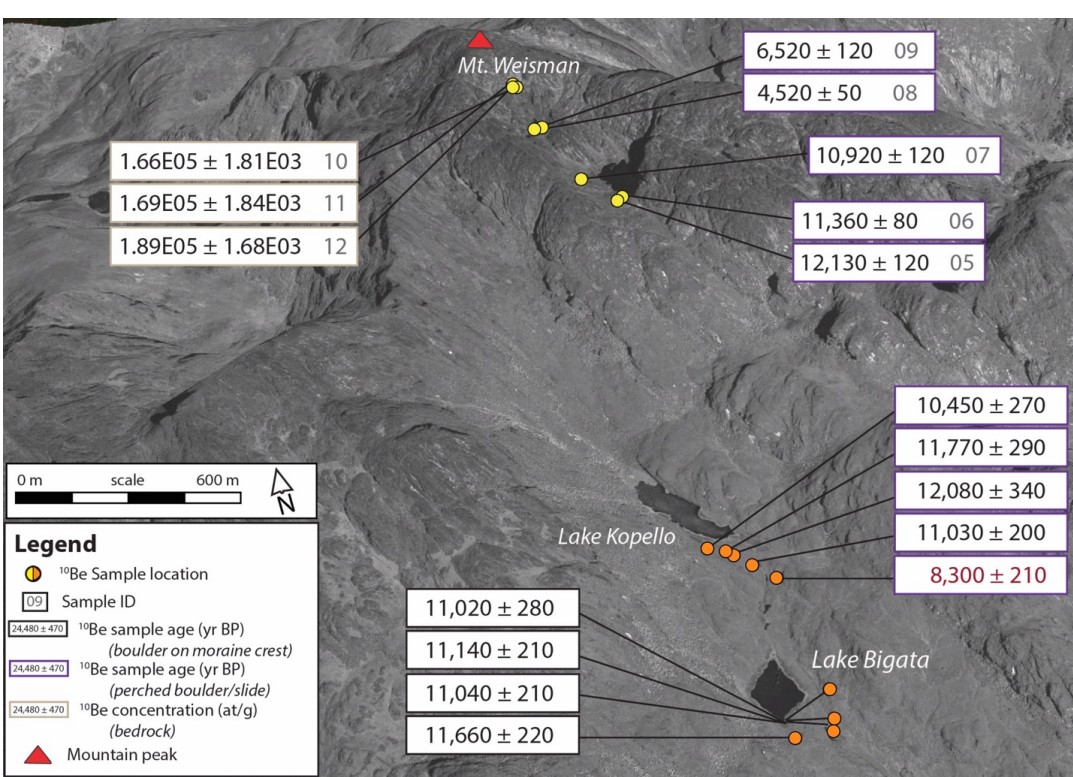

**Figure 4.** [10]Be ages and concentrations in the Nyamugasani valley. Ages are shown in black with internal analytical
uncertainties. Sample ID numbers as in Tables 1-3 are shown in grey. Sample locations are marked by yellow circles
and mapped onto a 0.5-m resolution Worldview-1 satellite image. Previously reported [10]Be ages from the moraine
that dams Lake Bigata and from boulders on bedrock near the outlet of Lake Kopello are marked with orange circles
(Jackson et al., in review). Ages considered to be outliers are shown in red.

**6 Discussion**
**6.1 Holocene Glacial Fluctuations in the Rwenzori Mountains**
The [10]Be ages from the Bujuku and Nyamugasani valleys suggest that glaciers in both
catchments retreated rapidly during the Early Holocene. Based on the lack of glacial deposits in
the Bujuku valley between the ~11.7 ka moraine and the Speke moraine, and the ~11.0 ka age of
the landslide that dams Lake Bujuku (Cavagnaro, 2017), we suggest that the former Bujuku
valley glacier retreated up the valley at, or shortly after, the onset of the Holocene. The landslide
occurs ~1.5 km up valley of the ~11.7 ka moraine and is undisturbed, indicating that ice had



retreated at least this distance up valley by ~11 ka and that ice remained up valley of this site
throughout the Holocene. Although it is possible that wetland or colluvium deposits on the valley
floor buried additional glacial deposits, there are no lateral moraines higher on the valley walls
and no evidence of glacial readvance over these colluvial sediments.

The morphology of the Speke moraine in the Bujuku valley, with steep ice-contact slopes

and more gentle ice-distal slopes (Figure 2), indicates that the lateral ridges formed as rock-fall
debris from the slopes of Mt. Speke fell onto the former Speke Glacier surface and was
transported supraglacially to the ice margin. The deposition of this rock-fall debris along the
glacier margins produced the fan-like ice-distal slopes of the Speke moraine. Deposition ceased
when the glacier receded and the moraine was abandoned at ~270-460 years ago.

Additional evidence for Early Holocene glacial recession in the Rwenzori comes from the

Nyamugasani valley. The stratigraphically innermost moraine (i.e., farthest up valley) in the
Nyamugasani valley dates to ~11.2 ka (Jackson et al., in review) (Figure 4). Approximately 0.5
km farther up valley, four boulders on a bedrock rise that dams Lake Kopello yield ages between
~12.1 and 10.5 ka (Jackson et al., in review). An additional ~1.6 km up valley from the Lake
Kopello bedrock, glacially-transported boulders set down on bedrock yield ages between ~12.1
and 10.9 ka. Based on the statistical similarity of these sample ages, we interpret these samples
to reflect rapid glacier recession from the innermost Nyamugasani valley moraine (~11.2 ka)
during the Early Holocene.  These data suggest that the valley was deglaciated to an elevation of
~4,430 m asl by at least ~10.9 ka.

Farther up valley, bedrock on the floor of Thomson cirque, below the south-facing peak

of Mt. Weisman, has $^{10}$Be concentrations equivalent to ~5.0-5.7 ka of net exposure (Figure 4).
The cirque was occupied by the former Thomson Glacier until at least the mid 20[th] century





(Meader, 1937; Whittow et al., 1963; Osmaston and Pasteur, 1972). This implies that the cirque
was ice-free for some period of time before its occupation by Thomson Glacier (Doughty et al.,
in press). If we assume that ice cover during the LGM (Kelly et al., 2014; Jackson et al., 2019)
was erosive enough to remove any pre-existing $^{10}$Be from the bedrock surface, the measured
bedrock $^{10}$Be concentrations reflect the total period of exposure of the bedrock (i.e., ice-free
conditions) after the LGM. More specifically, based on the Early Holocene age of moraines
down valley (Jackson et al., in review), we suggest the Thomson cirque bedrock $^{10}$Be
concentrations indicate the net duration of bedrock exposure (~5.0-5.7 ka) during the Holocene.

This interpretation, and the observation that the cirque was occupied by Thomson Glacier

during the early and middle 20[th] century, leads to one notable consequence. Namely, if the
Thomson cirque was ice-free for ~5.0-5.7 kyr during the Holocene yet occupied by ice during at
least a portion of Late Holocene time, this implies that ice had ablated away completely in the
cirque at some point earlier in the Holocene before re-nucleating prior to the 20[th] century. This
scenario may include multiple periods of glacial ablation and readvance, or a single period of
ice-free conditions followed by Late Holocene re-nucleation. Although the timing of ice
recession and re-nucleation within the cirque cannot be established with the data presented here,
the bedrock $^{10}$Be concentrations suggest that the cirque remained ice-free for a significant portion
of the Holocene Epoch.

The two boulders on bedrock ~10-20 m downslope of the cirque (dated to ~4.5 and 6.5

ka; RZ-15-07, 08) suggest that the former Thompson Glacier extended to this downslope
location during the Middle Holocene. However, the $^{10}$Be ages of these boulders are similar to the
exposure-age equivalent of the nearby cirque bedrock (~5.0-5.7 ka), which suggests that the
boulders may contain inherited $^{10}$Be. In this scenario, the boulders would have been plucked





from the nearby cirque floor and transported a short distance (~100 m) by the former Thomson
Glacier during a Late Holocene readvance. Alternatively, the boulders may have fallen onto the
ice surface from the valley walls above and escaped sub-glacial erosion prior to deposition.
Sample RZ-15-08 (~6.5 ka) is from an angular boulder, which may indicate that it was
transported supraglacially after falling onto the former ice surface from the valley headwall. In
contrast, sample RZ-15-07 (~4.5 ka) was sub-rounded in appearance and so was presumably
eroded during glacial transport. These sampled boulders are located near the early-20th century
snow or ice margin, as shown in photographs from 1937 CE (Meader, 1937). The proximity of
the samples to the early 20th century snow/ice margin supports the interpretation that the boulders
were deposited during a Late Holocene advance of Thomson Glacier, but more data are needed
to determine the depositional histories of these samples. Due to the uncertainties associated with
these samples, we do not use their $^{10}$Be ages in any subsequent interpretations.

Overall, $^{10}$Be ages from the Bujuku and Nyamugasani valleys suggest glacial recession

occurred in both catchments during the Early Holocene and that glaciers in these valleys then
retreated to, or inboard of, their maximum late-19th or early-20th century extents (Figure 5). In the
Nyamugasani valley, $^{10}$Be concentrations measured in bedrock samples from the floor of
Thomson cirque suggest ~5-6 kyr of net exposure of surfaces that were glacierized in the first
half of the 20th century. The results do not preclude the possibility that glaciers persisted on the
high Rwenzori peaks throughout the Holocene, albeit inboard of their late 19th century positions.
However, the data suggest that glaciers in the Bujuku and Nyamugasani valleys did not advance
beyond their Late Holocene maximum ice positions during Early or Middle Holocene time.

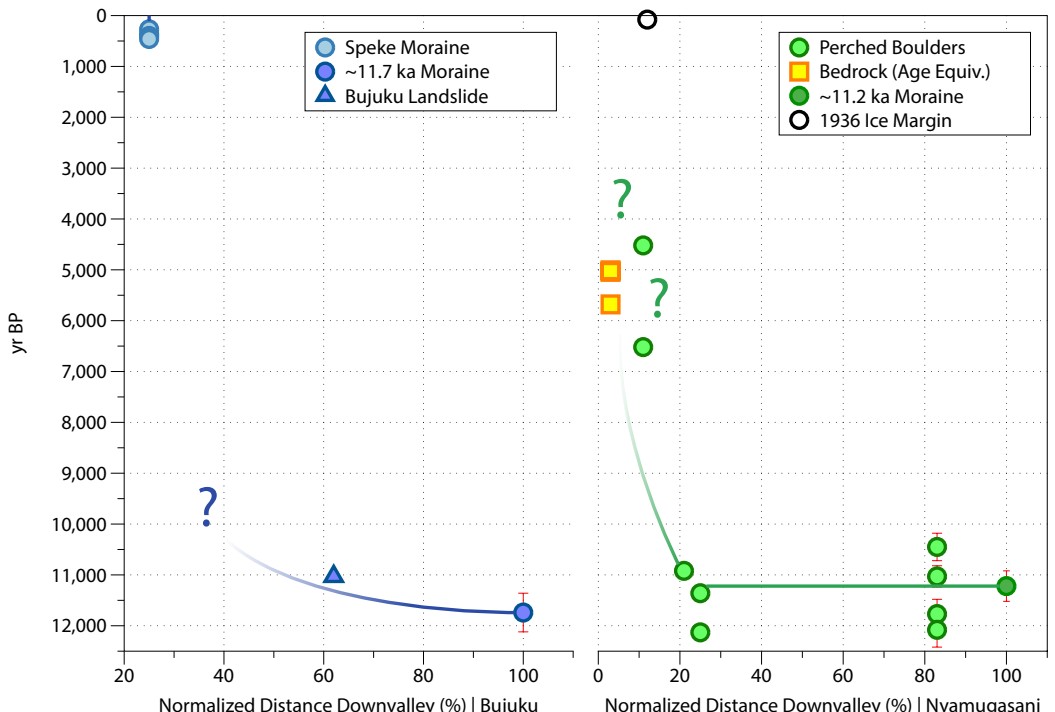


**Figure 5.** Normalized glacial extent in the Bujuku and Nyamugasani valleys during the Holocene. Lines indicate reconstructed glacial extent, question marks highlight periods where glacial extent is uncertain. (left) In the Bujuku valley, the maximum dated Holocene position (~11.7 ka moraine; Jackson et al., in review) is roughly 1.5 km down valley of the ~11 ka landslide which dams Lake Bujuku (Cavagnaro, 2017) and over 3 km down valley from the Speke moraine. (right) In the Nyamugasani valley, the most up valley moraine in the catchment dates to ~11.2 ka. Boulders set down on the Kopello ridge yield ages between ~12.1-10.5 ka. The Nyamugasani transect boulders yield similar ages to those on the Kopello ridge, but are 1.8 km farther up valley. Yellow squares mark the exposure-age equivalents ([10]Be concentration) of samples of bedrock from Thomson cirque, although we emphasize that these ages represent the cumulative exposure duration (likely since the LGM), and do not necessarily reflect the most recent period of exposure.

**6.2 Patterns of East African Glaciation and Temperature during the Holocene**

Although sensitive to precipitation, humidity, aspect, and hypsometry, glaciers in the 'humid' inner tropics (~10ºN-10ºS) are influenced primarily by temperature (e.g., Sagredo et al., 2014). This includes glaciers in the Rwenzori (Taylor et al., 2006; Russell et al., 2009; Kelly et al., 2014; Doughty et al., in press). The hypothesis that glacial fluctuations in the Rwenzori during the Holocene are controlled by temperature is supported by a comparison of our data with



paleolimnological records of regional precipitation and temperature (Figure 6). Organic
geochemical (branched glycerol dialkyl glycerol tetraethers; brGDGTs) proxy temperature
records from four East African lakes and the Congo River Basin indicate temperatures warmed
by ~1 ºC from ~12-11 ka, were similar to Late Holocene values from ~11 to 8 ka, and then rose
to a mid-Holocene thermal maximum between 6 and 5 ka, before cooling to Late Holocene
values (Ivory et al., 2017, and references therein). The Early Holocene is also marked by the
African Humid Period (AHP; ~11.6-5.0 ka), a time of elevated precipitation across tropical
Africa (Garcin et al., 2007) reflected in precipitation reconstructions from Lakes Victoria and
Tanganyika, the Nile River Delta, and at the foot of the Rwenzori Mountains at Lake Edward
(Russell et al., 2003a; Buening and Russell, 2004; Tierney et al., 2008; Berke et al., 2012;
Weldeab et al., 2014). Declining precipitation associated with the end of the AHP began at ~5.2
ka in western Uganda, recorded by rising salinity in Lake Edward (Russell et al., 2003b, Russell
and Johnson, 2006), roughly coincident with the onset of cooling in East Africa. Rwenzori
glaciers thus retreated during a wet and warming Early Holocene and remained near or inboard
of their Late Holocene maxima during the warm, drying Middle Holocene. The end of the AHP
and the onset of cooler conditions in East Africa broadly coincides with the transition to more
erosive glacial margins on Mt. Kenya (Karlen at el., 1999) and with the beginning of extended
net accumulation on the Kilimanjaro Ice Cap after ~4 ka (Gabrielli et al., 2014). We suggest that
regional temperatures were sufficiently high during the Early and Middle Holocene to dominate
glacial mass balance in the African tropics in spite of elevated precipitation.

The pattern of Holocene glacial fluctuations inferred in the Rwenzori is broadly

consistent with reconstructed glacial histories from elsewhere in East Africa. In Ethiopia,
Uganda, and Kenya, glaciers retreated either during or prior to the Early Holocene (Hamilton and





Perrot, 1982; Shanahan and Zreda, 2000; Tiercelin et al., 2008) and remained near or inboard of
reconstructed Late Holocene positions throughout Holocene time. Although disputed (Mahaney,
1989), there is some evidence for a Middle Holocene readvance of glaciers on Mt. Kenya
(Perrot, 1982; Johansson and Holmgren, 1985; Karlen et al., 1999). The ages of ~4.5-6.5 ka of
perched boulders in the Rwenzori's Nyamugasani valley may indicate a similar Middle Holocene
readvance, but to a position near or inboard of the maximum Late Holocene extent (Meader,
1937; Osmaston and Pasteur, 1972). Acknowledging this uncertainty, we suggest that the
Rwenzori chronology is generally representative of Holocene glacial fluctuations in tropical
Africa.

Our comparison of Rwenzori glacier extents with regional GDGT-based temperature

records indicates that ice masses did not respond linearly to temperature. For example, GDGT
temperature reconstructions suggest regional temperatures at ~11 ka were similar to temperatures
at ~1 ka-present (Ivory et al., 2017). In contrast, glacial margins during the Early Holocene were
~330 m lower in the Nyamugasani valley and ~490 m lower in the Bujuku valley than during the
Late Holocene (Figure 6). This difference may be due to the fact that there was more substantial,
if retreating, ice volume in the Rwenzori at ~11 ka relative to the Late Holocene, and that Late
Holocene ice was re-nucleating or re-advancing after a period of sustained ablation.
Alternatively, this difference may reflect two distinct equilibrium glacial mass balances at
similar temperatures but with different precipitation regimes and radiative boundary conditions.
Modeling suggests that past changes in Rwenzori equilibrium-line altitude are only weakly
influenced by precipitation amount, and that the large (~60% increase; Buening and Russell,
2004) changes in precipitation in western Uganda during the AHP are insufficient to explain the
large downslope movement of Rwenzori glaciers observed at ~11 ka (Doughty et al., in press).



Early Holocene glacial recession is coincident with both increasing atmospheric $CO_2$ (Monnin et
al., 2001)(Figure 6), which increases surface longwave radiation, and with rising mean-annual
equatorial insolation after ~10 ka (Berger and Loutre, 1991). Because tropical glaciers undergo
ablation throughout the year, mean-annual radiation (both insolation and longwave) influences
glacial mass balance in the tropics (Kaser and Osmaston, 2002) and may have played a role in
Holocene glacial extents. However glaciers in the Rwenzori and elsewhere in East Africa
apparently re-nucleated or readvanced during the Late Holocene, whereas atmospheric $CO_2$ and
mean annual insolation continued to rise. Alternatively, seasonal, rather than mean-annual,
insolation is another potential forcing mechanism for tropical glacial fluctuations. The sun passes
directly over the equator twice each year in September and March, coincident with the twice-
annual equatorial wet season (Sept.-Nov. and March-May) and the passage of the Intertropical
Convergence Zone over the equator (Singarayer and Burrough, 2015). Modeling studies of ice
cliffs on Kilimanjaro (Mölg et al., 2003a) and of modern Rwenzori glaciers (Mölg et al., 2003b)
suggest that a transition to more regionally arid conditions after ~1880 CE reduced cloud cover
and increased the amount of net annual solar radiation impacting glacier surfaces during dry
seasons, which may have encouraged further ablation after the initiation of recent Rwenzori
deglaciation ~1870 CE (Russell et al., 2009). Neither of the Rwenzori dry seasons (June-Aug
and Dec-Feb) had insolation minima during Middle Holocene (Figure 6), and so the Late
Holocene re-nucleation or readvance of African glaciers is difficult to reconcile with this
mechanism. Elevated wet season (Sept.-Nov.) insolation could play a role, but Sept. and March
insolation trends counteract each other during the Holocene. More work is needed to determine
the discrete influences on glacial mass balance over millennial timescales during the Holocene in
the East African tropics.





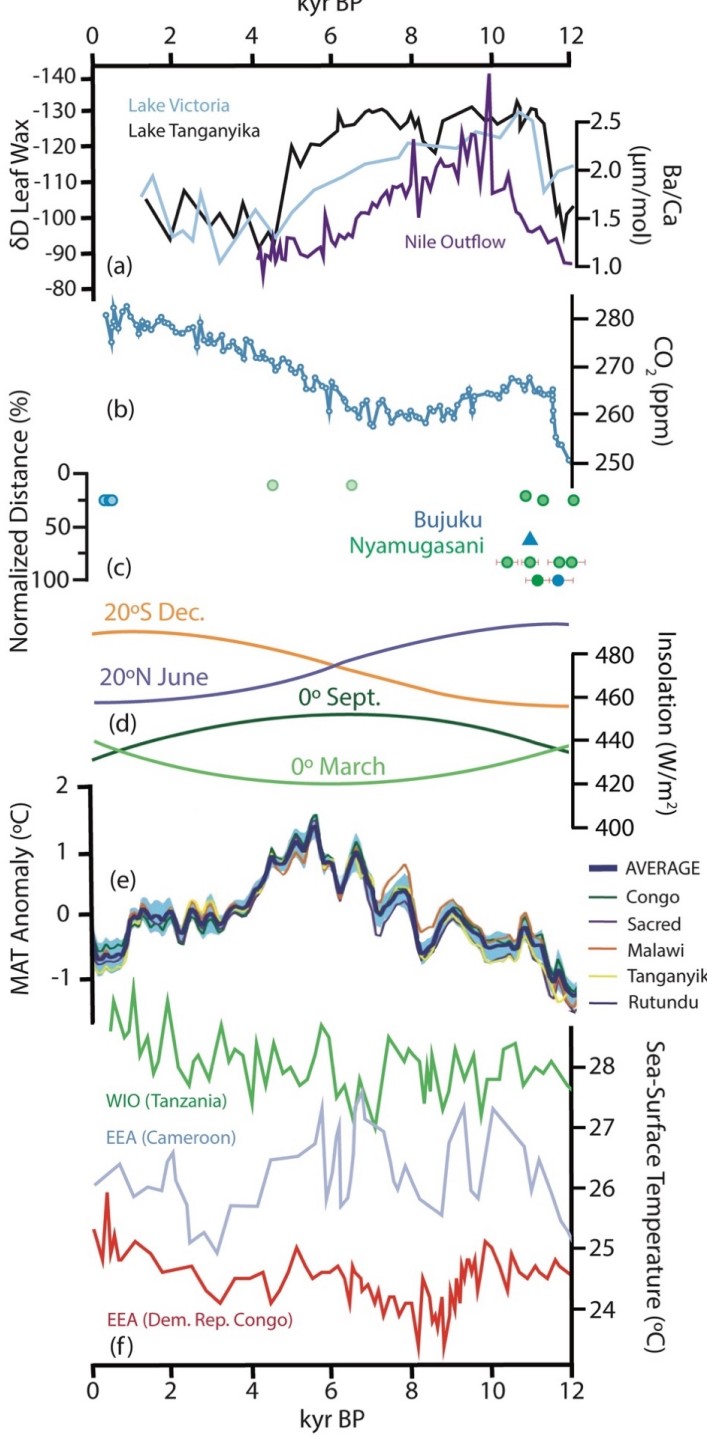

**Figure 6.** East African climate during the Holocene. (a) Precipitation (δD leaf wax) records from Lakes Victoria (light blue) and Tanganyika (black) and East Asian monsoon intensity (purple; Ba/Ca) from the Nile Delta (Tierney et al., 2008; Berke et al., 2012; Weldeab et al., 2014); (b) Atmospheric $CO_2$ (Monnin et al., 2001); (c) Normalized glacier distance down valley in the Nyamugasani (green) and Bujuku (blue) valleys in the Rwenzori Mountains as in Figure 5.; (d) Tropical Holocene insolation at 20°N (June; blue), 20° S (Dec.; orange), and °0 (Sept. (dark green) and March (light green) (Berger and Loutre, 1991); (e) Holocene terrestrial temperatures reconstructed from organic lacustrine sediments, bootstrapped and plotted as anomaly using the mean value over the last 2 ka (Ivory et al., 2017); (f) Sea-surface temperature records from the equatorial Western Indian Ocean (WIO; green; Romahn et al., 2014) and the Eastern Atlantic (EEA; light blue, red; Weldeab et al., 2005).

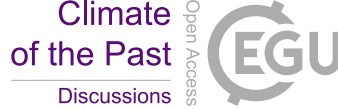

### 6.3 Tropical South American Glacial Fluctuations and Implications for Holocene Climate


Reconstructions of Holocene glacial fluctuations in South America show broad
similarities in the timing and magnitude of tropical Andean glacial extent change with records of
glacial extent from tropical Africa. Glaciers in the northern and southern tropical Andes retreated
during the Early Holocene and remained near or within their Late Holocene extents throughout
much of the Holocene Epoch (e.g., Jomelli et al., 2014; Stansell et al., 2017). [10]Be dating of Late
Holocene moraines documents glacial advances in the South American tropics after ~2-1 ka
(Solomina et al., 2015), and many glaciers only reached their Late Holocene maximum extents
within the last ~700-500 years (Licciardi et al., 2009; Jomelli et al., 2011; Jomelli et al., 2014;
Stansell et al., 2015; 2017). Sediment-flux analyses and radiocarbon dating of glacially
influenced lake sediments, however, indicate that glaciers were more erosive, and perhaps more
extensive, after ~5 cal kyr BP relative to earlier Holocene time (e.g., Rodbell et al., 2008). Lake
sediment records from sites in the Eastern Cordillera and Cordillera Blanca of Peru suggest that
glaciers in these regions advanced and retreated multiple times during the Holocene, with more
advanced ice positions taking hold after ~4-2 cal kyr BP (Rodbell et al., 2008; Stansell et al.,
2015; 2017). [10]Be dating and geomorphic mapping of moraines from these sites also suggest that
glaciers generally remained inboard of their Late Holocene maxima until at least ~1 ka (Stansell
et al., 2015; 2017). Altogether these data suggest that, after a period of Early Holocene retreat,
glacial margins in the South American tropics were more dynamic after ~ 5 ka relative to the
Early Holocene but did not achieve their maximum extents until after ~1 ka.
The broad similarity of Holocene glacial fluctuations in tropical East Africa and South
America suggests that tropical glaciers responded to a common, pan-tropical forcing mechanism
during the Holocene. Based upon prior observational and modeling work assessing controls on





glacial mass balance across the South American Cordillera (Sagredo and Lowell, 2012; Sagredo
et al., 2014), we suggest that temperature was the primary control on glacial extents across the
low latitudes during the Holocene. This hypothesis requires that Holocene temperatures were
similar across the tropics, and that whatever mechanism or mechanisms affected temperatures in
one region had similar impact elsewhere. As in East Africa, radiative forcing from atmospheric
$CO_2$ and mean-annual or seasonal equatorial insolation cannot easily explain the pattern of
glacial fluctuations in the South American tropics. This highlights an avenue for future research
through both ground-based geologic investigation as well as through climate and mass-balance
modeling. Determining the mechanisms that influence temperatures in the low latitudes is crucial
for understanding better the context for modern warming and the sensitivity of tropical glaciers
to future climate change.

**7 Conclusions**

Twelve new [10]Be ages of glacial features in the Rwenzori Mountains indicate that glaciers

retreated rapidly during the Early Holocene and remained near or within their Late Holocene
extents through much of the Holocene Epoch. These results are broadly similar to records of past
glacial fluctuations elsewhere in tropical East Africa. Based on a comparison of tropical East
African glacial fluctuations with regional climate records, we suggest that temperature acted as
the primary control on glacial fluctuations throughout the Holocene.

Glacial chronologies from tropical Africa and South America indicate that Early

Holocene glacial recession was followed by a period of generally restricted ice extents until at
least ~ 1 ka. The coherence of tropical African and South American glacial fluctuations suggests
that glaciers across the low latitudes responded to a common forcing during the Holocene, which



we suggest was most likely temperature. However the ultimate driver of Holocene temperatures,
and thus glacial extent, remains enigmatic. Understanding the controls on low-latitude
temperature is crucial for assessing and contextualizing modern climate variability and for
determining the sensitivity of tropical glaciers to changing climate conditions. Although more
work is needed to assess the sensitivity of low-latitude temperature to discrete forcing
mechanisms, the results presented here highlight the utility of glacial records in assessing past
terrestrial temperature change in tropical regions.


















**Data Availability**

All analytical information and metadata associated with newly reported cosmogenic nuclide

measurements are included within the manuscript tables (Tables 1-3). All reported cosmogenic

nuclide ages are as calculated using the ICE-D calibration database (calibration.ice-d.org) and

version 3 of the online exposure-age calculator as described by Balco et al., 2008 and

subsequently updated (hess.ess.washington.edu).

**Author Contribution**

MK, JR, and AD designed the project. BN coordinated the project in Uganda. MJ, AD, JR, and

MK collected samples. MJ and JH processed samples for [10]Be dating and SRHZ measured

beryllium ratios. MJ, MK, AD, and JR analyzed results. MJ wrote the paper with contributions

from all authors.

**Competing Interests**

The authors declare that they have no competing interests.

**Acknowledgements**

We thank the Uganda Wildlife Authority and the Uganda National Council on Science and

Technology for their support of and assistance with this project. We also thank Rwenzori

Mountaineering Services and Rwenzori Trekking Services for their logistical support while in

the field. Laura Hutchinson and David Cavagnaro helped process samples. This project was

supported by the National Science Foundation (EAR-1702293; GSS-1558358), Comer Science

and Education Foundation (CP103), Comer Global Climate Change Foundation (GCCF13), and





Sigma Xi (G20141015664263). Satellite imagery was granted by the DigitalGlobe Foundation
(0054508271). This is LLNL-JRNL-807478.























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
