# Peer review of "Holocene glaciation in the Rwenzori Mountains, Uganda"

_Climate of the Past, 2020_

## Referee Comment (RC1) · Anonymous Referee #1 · 21 May 2020

Decision: reject – inappropriate journal.

The manuscript "Holocene glaciation in the Rwenzori Mountains, Uganda" by Jackson et al., presents a new set of Holocene cosmogenic dates (n=12) from 2 valleys in Uganda. In my opinion, this study should not be published in the journal Climate of the Past. The primary reasons for this are (1) it is too regional in significance to be appropriate for COP (note the title- does not really reflect an inference about climate); (2) in my opinion it has a very limited ability to provide any concrete climate inferences, not because of the study design, but because of the number of samples, and the quality of the data that can be provided from the limited chronological constraint provided by these dates.; (3) the main conclusions, while theoretically plausible, are not unambiguously supported by the data, and this is not clear from the abstract or conclusions; (4)

I am concerned that given the reliance on ages that are presented and under review elsewhere (Jackson et al., under review), for the interpretations that are being made in this paper, that really the two papers should have been combined and splitting the manuscripts in two seems unjustified. I explain in more detail below.

The paper presents only 12 dates – a small number to constrain any sort of cosmogenic glacial history, particularly since the authors have several other papers published or currently in review from the same sites. Without seeing the other papers, the apparent justification is that this paper is focused on Holocene variability. The authors thus present some older deglacial ages (e.g., 10-12 kyr BP) that appear to be from the other Jackson et al.,in press paper, some latest Holocene ages (300-500 yr BP, n=4), in another valley 5 boulders on bedrock – not associated with glacial moraines (11-12 kyr (n=2), ~4 kyr BP n=3) and samples of bedrock in the uppermost cirque from this valley (5-6 kyr, n=3).

The way I would interpret this data is that it does seem that you had an early Holocene deglacial retreat at 11-10 kyr BP. And there is evidence from one site that there was a small standstill or readvance during the Little Ice Age (note that this readvance is seen in other African localities, including Mt Kenya and Kilimajaro, I think. The interpretation of the other boulder ages is ambiguous. The upvalley cirque ages of 5 kyr are the same as the valley boulders. So how do we interpret these boulder ages? They are not associated with any geomorphic features, so they may just reflect material deposited during retreat of the ice, and their age may not have any real meaning (ie perhaps they are simply inherited cosmogenic nuclides. Alternatively, the 5kyr ages in the uppermost cirque may suggest that the valleys were basically ice free by 5 kyr.

From this data they make several inferences:

1- glaciers did not readvance beyond their late Holocene maxima during the early or mid Holocene. This is possible, but given that the record is inherently erosive, it is hard to say anything from the absence of evidence. The old ages in the upper cirque

bedrock do seem to support this inference (that ice was gone by 5 kyr) but then the authors also say this: Line 355 "Although the timing of ice recession and re-nucleation within the cirque cannot be established with the data presented here, the bedrock 10Be concentrations suggest that the cirque remained ice-free for a significant portion of the Holocene Epoch."

2- Line 436 "our comparison. . . indicates that ice masses did not respond linearly to temperature." From the really limited data here, I don't see how you can say this. The delay may be simply a lag in the response rather than a nonlinear adjustment

3- Line 552 conclusion: "Based on a comparison of tropical East African glacial fluctuations with regional climate records, we suggest that temperature acted as the primary control on glacial fluctuations throughout the Holocene"

I have two concerns about this statement. First is that it is hard to establish the relationship between the glacial chronology and temperature reconstructions given the limited constraints. Essentially what we have is that ice retreat started in the early Holocene and was likely complete by 5 kyr (unless there is cosmogenic inheritance in the upper cirque bedrock). The African lake temperature records do show a signal that is consistent with this but it is not the only explanation. For example, several of the precipitation records also show declining values from 11-12 kyr and could be partially responsible for the glacier retreat. Essentially the problem is that we have very little we can say about what happened during the Holocene from these samples. Furthermore, there is no apparent cooling during the Little Ice Age in the temperature records which could account for the glacial readvance at that time.

Its not clear that the South American comparison is all that convincing of a tropics-wide temperature mechanism. Both Africa and South America chronologies suggest that retreat started early and the most extensive subsequent advance was in the latest Holocene. But South American records show evidence of much greater dynamics (presumably associated with rapid retreat) after 5 kyr, whereas at Ruwenzori, the ice

was already nearly gone by that time.

As a final point, I note that the authors did not appropriately reference the literature in making some of their statements. For example:

Abstract Line27: I think it is incorrect to say that "little is known about the response of tropical glaciers". There is literature from South America for certain, and though I am less familiar, probably in Asia as well.

L411 –Garcin 2007 is not the appropriate reference for the African Humid Period. I suggest referencing some of the early primary literature by Francoise Gasse and other leaders in the field.

---

## Short Comment (SC1) · 6 Jun 2020

Jackson et al. provide new evidence for Holocene glacier fluctuations in tropical Eastern Africa based on 10Be surface exposure dating of moraine boulders (n = 4), unassigned boulders (n = 5), and bedrock (n = 3). Glacial deposits are a valuable proxy for past climatic and environmental changes and often constitute the only possibility for the direct dating and reconstruction of these changes, especially at high elevations. To better understand the impact of past climatic changes on the afro-alpine environment and, vice versa, to infer potential implications of past glacier fluctuations for the tropical palaeoclimate, new insights into the Holocene glacial history of Eastern Africa are most welcome. However, I think it would be worthwhile to specify the aim of the study and reconsider some of the general conclusions, as I will further outline below.

General comments:

Following previous publications (Kelly et al., 2014; Jackson et al., 2019; Jackson et al., in review), this manuscript is the fourth one presenting glacial chronological data from more or less similar sites in the Rwenzori Mountains. Splitting a glacial chronology into several papers might be reasonable to have more space for discussing individual aspects/events in detail, but the added value of this manuscript remains unclear after first reading. Most of the exposure ages from the Nyamugasani Valley (Fig. 4) that determine the glacier extent at ∼12-11 ka (before the onset of deglaciation) stem apparently from the other manuscript in review (Jackson et al., in review). From my understanding, the "only" new finding based on the additional ages from the upper part of the valley is that the "Thomson cirque" (and maybe Mount Weisman too?) was probably ice-free by ∼5 ka or at least for a longer period during the Holocene. Since the reader has no insight into the other manuscript in review, it would be important to elucidate the novelty or new aspect of the contribution presented here.

In the abstract, the authors propose that "understanding how tropical glaciers responded to past periods of warming is crucial for predicting and adapting to future climate change [...]" (lines 26-27). They state further in the introduction that "tropical glaciers are a primary source of freshwater and are a fundamental component of regional economies [...]" (lines 52-53) and that "determining when and how glaciers in the African tropics fluctuated during past warm periods provides crucial information for assessing whether, or how long, tropical glaciers may persist under future warming scenarios" (lines 70-72). Although the ongoing glacial melting in the tropics and most of the mountains worldwide is of great concern, the contribution of the meltwater from the relatively small glaciers in equatorial Eastern Africa to the alpine runoff is negligible (e.g. Kaser et al., 2004; Taylor et al., 2009) and thus do not seem to play a major role for the regional economy and freshwater supply. Moreover, the authors do not explain how limited information on past glacial fluctuations could help to better project the future evolution of tropical glaciers in response to global climate change. Reconstructed

glacier extents and established glacial chronologies provide without doubt important information on past glacier dynamics, but I think the palaeoclimatic, -environmental, and -glacial data are too uncertain to draw meaningful conclusions about "[. . .] the sensitivity of tropical glaciers to future climate change" (lines 545-546). I would even argue the other way round that modern observations and investigations regarding the climate sensitivity of tropical glaciers in Africa are inevitable for a reliable interpretation of past glacier fluctuations in the region (e.g. Mölg et al., 2003; Mölg et al., 2004; Mölg et al., 2008; Mölg et al., 2009; Nicholson et al., 2013). It is a bit surprising that the authors do not pick up the topic again in the discussion and do not emphasize the claimed relevance of their findings for the future evolution of tropical glaciers. I would therefore recommend that the authors rather stress the palaeoclimatic and -environmental relevance of their study in the abstract and introduction.

The two main conclusions of the manuscript are that (1) Holocene glacier fluctuations were similar across the tropics and based on the consideration of regional climate records that (2) "[. . .] temperature acted as the primary control on glacial fluctuations throughout the Holocene" (lines 553-554). I do not agree with these statements for the following reasons:

1. The 10Be exposure ages from the Holocene moraine stages in the Rwenzori Mountains (Nyamugasani Valley), on Mount Kenya (Teleki Valley), and on Kilimanjaro (Kibo Peak) originate more or less from one valley/locality (Shanahan and Zreda, 2000). Whether the respective ages are representative for the entire mountain range can neither be confirmed nor refuted. I think without further evidence it remains hypothetical whether glaciers in tropical Eastern Africa responded synchronously to Holocene climate changes or not.

2. The Early and Middle Holocene moraine stages dated in the Rwenzori Mountains (∼11.7 ka), on Mount Kenya (∼10.2 and ∼8.6 ka), and on Kilimanjaro (∼13.8 ka) show by no means a similar pattern, apart from a general warming trend after the last glacial period. The differences could be explained by dating uncertainties, but also by climatic

variations. How do you interpret the differences?

3. In view of a lacking robust Holocene glacial chronology for Eastern Africa, the dynamic Holocene glacier fluctuations in South America, and the non-consideration of other tropical glacial chronologies, claiming "[similar] Holocene glacial fluctuations across the tropics" (lines 38-39) seems rather speculative than evidence-based. Moreover, this assumption underrates the complex regional response of alpine glacier to climatic changes (e.g. variations in temperature, precipitation, cloudiness, insolation, and moisture) in general. Mountain height, terrain, hypsometry, debris cover, glacier size, and many other geological, geomorphological, glaciological, and climatic parameters control the magnitude and rate of glacier fluctuations, as the regional variations in the response of alpine glaciers to recent global warming underline (e.g. Zemp et al., 2019).

4. A key assumption for the author's hypothesis that " [. . .] tropical glaciers responded to a common, pan-tropical forcing mechanism during the Holocene" (lines 534-535) is that tropical glaciers are highly sensitive to changes in temperature (lines 399-404; see also Jackson et al., 2019). As a reference for the Rwenzori Mountains, the authors quote a controversial study (Taylor et al., 2006a; Taylor et al., 2006b) which claims that rising temperatures are the dominant factor for recent glacier melting in the Rwenzori Mountains. However, the detailed comment on this study by Mölg et al. (2006), which elaborates the importance of other climate variables for the energy and mass balance of tropical glaciers, is neglected in the discussion. Multiple studies from Kilimanjaro and the Rwenzori Mountains stress that climate variables related to air moisture (e.g. specific humidity affecting sublimation, cloudiness affecting incoming solar radiation, precipitation affecting glacier surface albedo and mass gain) play an important role in the present surface energy balance of tropical glaciers in Eastern Africa, especially at high elevations above the 0 °C isotherm (Mölg et al., 2003; Mölg et al., 2004; Mölg et al., 2006; Mölg et al., 2008; Mölg et al., 2009; Nicholson et al., 2013). Since the sensitivity of tropical glaciers in Eastern Africa to different climate variables is an ongoing and very important debate that is crucial for the hypothesis and conclusions of the presented manuscript, the controversial arguments should find more attention in the discussion. In view of the modern observations, I doubt that past glacial fluctuations in Eastern Africa can and should be explained by temperature variations alone.

Specific comments:

Fig. 1b: Would it be possible to add geographic coordinates to the map of the central Rwenzori massif?

Fig. 1b: For me as a reader who is not familiar with the region, it is difficult to interpret the terrain on the Worldview-1 satellite image. Replacing the image by a combination of DEM and hillshade (including the shapes of the lakes) might be an alternative.

Fig. 2: Could you include at least one photo of a sampled boulder and bedrock surface so that the reader gets a better impression of the investigated landforms?

Lines 135-138 and 425-427: The authors rely solely on radiocarbon ages from lake Garba Guracha in the Bale Mountains to discuss the potential timing of deglaciation in the southern Ethiopian Highlands, although direct 36Cl surface exposure ages of 21 moraine boulders from two valleys in the Bale Mountains are published (see Fig. 1 and S6-8 in Ossendorf et al., 2019). The inner-most moraines in the two valleys show that deglaciation in the Bale Mountains began after ∼15-14 ka and suggest (not necessarily imply) that the southern Ethiopian Highlands were ice-free before the Pleistocene-Holocene transition.

Lines 254-256.: Did you conduct a simple sensitivity analysis (assuming e.g. two or three plausible erosion rates) to assess the age uncertainty related to erosion?

Table 1: Content-wise, the columns with the 10Be concentrations in Table 1 would fit better in Table 2. Information of sample lithology could be included in Table 1 if available.

Table 3: Could you outline how you define the "internal" (probably analytical) and "external" error?

Fig. 4: I understand why you report 10Be concentrations instead of exposure ages for the bedrock samples (RZ-15-01, RZ-15-02, RZ-15-03), but they are difficult to interpret and compare with the other results. Therefore, I would recommend to report the exposure ages (instead of concentrations) in the map and note in the legend that they indicate the net duration of bedrock exposure, as you outlined in the text.

Fig. 4 and Table 3: Considering the general uncertainties associated with surface exposure dating (analytical errors, unknown erosion rates, etc.), I don't see justification to report ages in a way (e.g. 11,020 ± 280 years) that implies the method is precise enough to date events to a specific decade. I would recommend to round the ages and report them in kiloyears (e.g. 11.0 ± 0.3 ka).

Section 6.3: What is the rationale to explicitly discuss the glacial fluctuations in tropical South America here, although no new or recalculated ages are presented? The aim/motivation for the exclusive comparison between the Holocene ages from the Rwenzori Mountains and Andes is not clear from the abstract and introduction. Glacial chronological data also exist from other locations across the tropics.

References:

Jackson, M.S., Kelly, M.A., Russell, J.M., Doughty, A.M., Howley, J.A., Cavagnaro, D.B., Zimmerman, S.R.H., and Nakileza, B.: Glacial fluctuations in tropical Africa during the last glacial termination and implications for tropical climate following the Last Glacial Maximum. Quaternary Science Reviews (in review).

Jackson, M.S., Kelly, M.A., Russell, J.M., Doughty, A.M., Howley, J.A., Chipman, J.W., Cavagnaro, D., Nakileza, B. and Zimmerman, S.R.: High-latitude warming initiated the onset of the last deglaciation in the tropics. Science Advances 5, 1-8. 2019.

Kaser, G., Hardy, D.R., Mölg, T., Bradley, R.S., and Hyera, T.M.: Modern glacier retreat on Kilimanjaro as evidence of climate change: observations and facts. International

Journal of Climatology 24, 329-39. 2004.

Kelly, M.A., Russell, J.M., Baber, M.B., Howley, J.A., Loomis, S.E., Zimmerman, S., Nakileza, R., and Lukaye, J.: Expanded glaciers during a dry and cold Last Glacial Maximum in equatorial East Africa. Geology 42, 519-522. 2014.

Mölg, T., Georges, C., and Kaser, G.: The contribution of increased incoming short-wave radiation to the retreat of the Rwenzori glaciers, East Africa, during the 20th century. International Journal of Climatology 23, 291-303. 2003.

Mölg, T., and Hardy, D.R. : Ablation and associated energy balance of a horizontal glacier surface on Kilimanjaro. Journal of Geophysical Research 109, 1-13. 2004.

Mölg, T., Rott, H., Kaser, G., Fischer, A., and Cullen, N.J.: Comment on 'Recent glacial recession in the Rwenzori Mountains of East Africa due to rising air temperature' by Richard G. Taylor, Lucinda Mileham, Callist Tindimugaya, Abushen Majugu, Andrew Muwanga, and Bob Nakileza. Geophysical Research Letters 33, 1-4. 2006.

Mölg, T., Cullen, N.J., Hardy, D.R., Kaser, G., and Klok, L.: Mass balance of a slope glacier on Kilimanjaro and its sensitivity to climate. International Journal of Climatology 28, 881-892. 2008.

Mölg, T., Cullen, N.J., and Kaser, G.: Solar radiation, cloudiness and longwave radiation over low-latitude glaciers: implications for mass-balance modelling. Journal of Glaciology 55, 292-302. 2009.

Nicholson, L.I., Prinz, R., Mölg, T., and Kaser, G.: Micrometeorological conditions and surface mass and energy fluxes on Lewis Glacier, Mt Kenya, in relation to other tropical glaciers. The Cryosphere 7, 205-1225. 2013.

Ossendorf, G., Groos, A.R., Bromm, T., Tekelemariam, M.G., Glaser, B., Lesur, J., Schmidt, J., et al.: Middle Stone Age foragers resided in high elevations of the glaciated Bale Mountains, Ethiopia. Science 365, 583-587. 2019.

Taylor, R.G., Mileham, L., Tindimugaya, C., Majugu, A., Muwanga, A., and Nakileza, B.: Recent glacial recession in the Ruwenzori Mountains of East Africa due to rising air temperature. Geophysical Research Letters 33, 1-4. 2006a.

Taylor, R.G., Mileham, L., Tindimugaya, C., Majugu, A., Muwanga, A., and Nakileza, B.: Reply to Comment by T. Mölg et al. on 'Recent glacial recession in the Rwenzori Mountains of East Africa due to rising air temperature.' Geophysical Research Letters 33, 1-4. 2006b.

Taylor, R.G., Mileham, L., Tindimugaya, C., and Mwebembezi, L.:Recent glacial recession and its impact on alpine riverflow in the Rwenzori Mountains of Uganda. Journal of African Earth Sciences 55, 205-13. 2009.

Zemp, M., Huss, M., Thibert, E., Eckert, N., McNabb, R. Huber, J., Barandun, M. et al.: Global glacier mass changes and their contributions to sea-level rise from 1961 to 2016. Nature 568, 382-386. 2019.

---

## Referee Comment (RC2) · Anonymous Referee #2 · 17 Jun 2020

The paper by Jackson et al. provide twelve new 10Be based surface exposure ages from the Rwenzori Mountains of Uganda to place new constraints on late Pleistocene and Holocene glacial changes. Overall, I find the paper easy to read and think the results could be of interest to a broad audience such as the readership of Climate of the Past. However, I think the authors need to address several shortfalls before the paper is ready for publication. Below I have provided my line number edits and comments.

Abstract: - While I agree that understanding tropical glaciation in the past is a worthwhile endeavor, I think the authors have overstated how much we do not know about tropical glaciation over the Holocene. Certainly, there is more than "relatively little" known in my mind. I suggest trying to be more quantitative about what they are refer-

[Figure]

ring by "little" so the reader can better understand their argument.

Line 171, 199, etc.: The authors cite the work of Jackson et al. in review in multiple places within the paper. I am not sure about CPs policy on citing in review or un-published peer review work but from a reader's perspective this seems unusual and unhelpful since I cannot reference back to the paper to understand what is being cited. I also think the authors are relying quite a bit on this other data and wonder why the two papers have been separated from one another given these data from the in review paper are somewhat critical to the author's arguments.

Figure 2: It would be useful in Figure 2c if the authors provided some indication of the moraine crest for the reader – perhaps some arrows.

Figure 2: In Figure 2c there appear to be many trees within the photo. I'm confused by this photo given that the authors state on lines 250-252 that vegetation is sparse above 4000 m and that vegetation cover was not used to correct the 10Be dates. Some explanation seems warranted here since there appears to be clear signs of heavy vegetation cover in this region. Is this the ∼300-400 year Speke moraine, or is it an older moraine?

Lines 231: The process blank 9/10Be ratio should be provided as well as the carrier name or stock number. This is important for historical documentation.

Lines 238-240: The authors should say why they choose one over the other and they should say why it does not alter their conclusions. This would help the reader more fully understand their position on the matter.

Line 240-243: While I understand why presenting the ratios might be justified, I think it makes the several parts in the manuscript confusing for the reader doing it this way. Especially, when the authors then use the calibrated ages later on in the study (see the Discussion). I suggest providing the calculated ages and simply explaining why they are likely complex ages related to prior exposure.

Line 248-250: Provide a citation for the daytime temperatures and solar radiation that are being referenced. As-is, I find the snow correction argument weak and it needs more justification which I think a few references could help with.

Line 277: In the methods, the authors state that they use the LSDn scaling (Line 239) but throughout the text and starting here they seem to be using the St scaling. This needs to be corrected in the paper and/or table.

Line 288: It would be useful to the reader if the authors provided some photos of the boulder that they are referring or some dimensional information about the boulders. This is important information to convey to the reader since the classification of boulder is large (i.e. > 256 mm).

Figure 4: - I recommend providing ages instead of ratios for the three bedrock samples - In the legend and the boxes, it is hard to differentiate between the colors. Either makes these lines thicker and/or make the colors standout more between each other. - The sample ID provided in the boxes do not match the samples numbers from the tables. I'm therefore not sure what the samples numbers in the boxes represent and would suggest matching them to the tables for reader. - The authors provide data from an in review paper. I find these data quite helpful for their argument but unfortunately because it is in another paper I cannot reasonably evaluate the data. Again, I wonder why the authors have split the data between the papers but think until the in review paper is published it makes these data less convincing to the author's arguments.

Line 313: When the authors say "glacial deposits" do they also mean till? Is it bare bedrock?

Line 314: It would be useful to see a more zoomed out view of the Bujuku valley so the ~11 ka moraine vs. the Speke moraine can be seen.

Line 315: Cavagnaro, 2017 is a undergraduate thesis. I'm not sure if it is appropriate to cite work that is not peer reviewed.

Line 317: While the landslide is not disturbed, is it possible it occurred onto the glacier and then melted out? Without more information about the landslide and how it was dated, it is hard as a reader to evaluate if this is true without having a photo of the landslide or more information beyond the unpublished thesis that I was not able to access.

Line 322-325: Here I strongly disagree with the authors. Steep ice-contact slopes and more gentle distal slopes are the norm for moraines and especially true for young moraines that are ice cored and late Holocene in age. Therefore, I don't understand how the authors conclude the moraine was related to rock fall. Is the moraine highly sorted? Is it possible the "moraine" is in fact a protalus rampart (e.g. Ballntyne and Kirkbride, 1986)?

Line 326: Based on the imagery I cannot see the fan-like slopes. Is it possible to provide some zoomed in images of what is being referred?

Line 327: If the moraine is in fact a debris slide onto the slope, then how can the ages be interpreted as the timing of moraine abandonment? It seems like the boulder ages should be predating the timing of the glacier retreat. This needs to be explained.

Line 339: The ages are now being presented for the bedrock ages. The authors need to decide if they are going to present the ages or the ratios. I suggest the former.

Line 340: What evidence is this based on (photos, documentation, etc.)?

Line 357-358: This argument is reasonable but the assumption that LGM ice core erased all prior exposure needs to be restate here for the reader. Also, it would be worth providing some justification about why this assumption is more reasonable than inheritance being pervasive.

Lines 367: Again, some boulder photos would go a long way.

Line 373: The author might comment on why there is no late Holocene moraine at both locations? Is this an elevation or aspect issue?

Line 402: Again, it is hard to evaluate new and prior work that is not published yet.

Line 437: It is not clear what the authors mean. Presumably the glaciers respond to temperature in the same way – more temperature means more ablation and vice versa. I think other factors come into play here (hypsometry, winter precipitation, energy balance) and therefore it isn't that they respond non-linearly it is that other factors are important and that temperature is not the only driver of ice mass position.

Line 538-540: If the glaciers are similar tropic wide, then I would expect other climate archives to also reflect this. What about the PAGES 2k reconstructions or the Marcott et al. 2013 stacks? Do they show a similar pattern as suggested by the glacial data? These needs further justification.

---

## Referee Comment (RC3) · Anonymous Referee #3 · 26 Jun 2020

This short manuscript is clearly written and well-organized, and reports a small but interesting new set of twelve Be-10 ages from glacial moraines, boulders, and bedrock surfaces in the Rwenzori Mountains. The authors present some reasonable interpretations of the Holocene glacial history at their field sites based on their data and field observations. These results are then compared with regional climate proxies and other glacial records in East Africa, and also to tropical glacier records in South America. Overall, I think these newly reported findings from the Rwenzori are valuable and add important knowledge to the glacial and climate history of tropical East Africa. However, the number of new exposure ages is quite modest, and as such, it is difficult to extrapolate from these to make strong arguments about commonalities with Holocene glacier records in South America and pan-tropical climate forcings. I support the publication of

these results after appropriate revisions, but I urge the authors to be more cautious and realistic about the limitations of inferring global-scale correlations and climate forcing mechanisms from a small data set.

More specific comments and critiques are listed below. I hope the authors find these constructive, and I encourage them to address and resolve these in order to improve their manuscript.

Lines 27-28: That's probably not a fair statement these days, at least outside Africa, as there have been a number of studies and reviews of tropical glaciation in recent years.

Line 68: List in chronological order. It's also curious that the Late Holocene is not regarded here as a time period of interest - especially in light of statements about this work's relevance to modern/future climate change.

Line 82 / Figure 1: The satellite image in panel b is not an acceptable substitute for a proper glacial-geomorphic map. The moraines and overall topography are very hard to see. I suggest replacing with a DEM or contour map if available, overlain by a more detailed map indicating glacial features and other relevant landforms in these valleys. Without a well-labeled glacial-geomorphic map, the text descriptions of these field areas are very hard to follow.

Line 93: That is very inclusive. What kind of crystalline rock, exactly?

Line 111: This is the first of many citations to an in-review manuscript that is not currently accessible to reviewers. Because many interpretations here are reliant on context and support from the results in the unavailable in-review manuscript, it is not really possible to properly assess this new manuscript. In fact, the frequent references to the in-press manuscript and the importance of those findings to the interpretation of the new ages reported here raises the question of why these data were not all reported together in a single paper.

Line 123: I assume these calendar ages are recalibrated from the original radiocarbon

data using updated calibration curves, but the details of the calendar age estimation need to be explained here.

Line 190: How is the landslide dated? How reliable is that age?

Line 229 / Table 1: Density and erosion columns can be eliminated since the values are uniform for all samples. Instead, just note these values in a footnote. Also, how close are the three boulders (RZ-12-22, 24, 25) with indistinguishable latitude-longitude coordinates? A field photo would help show the field relations.

Lines 241-244: It's an odd choice to show Be-10 concentrations instead of apparent ages for the bedrock surfaces on Figure 4, even if there's a suspicion of complex exposure scenarios. This forces readers to find the ages in Table 3 (where they are reported) to gain some sense of the exposure durations and how they fit in with the other ages on the map. Also, if isotope inheritance is the concern, then that same issue could also potentially apply to the boulder surfaces - as acknowledged in the discussion.

Line 246 / Table 2: Why are the isotope ratios in a different table than the concentrations (and the ages, for that matter)? I suggest some consolidation of the three tables, ideally into one table if possible. The first three columns are identical in all of them, other columns can be eliminated (as noted earlier), and it's inconvenient to have to retrieve data from individual samples spread across three different tables. Also, given that the sample ratios are just over one order of magnitude above the blanks, it is important to consider how well these blank values are known. If they are all prepared from the same spike, it appears they vary quite a bit - and are therefore known with far less certainty than implied by the analytical uncertainties on individual blanks. This is a potentially important source of uncertainty for the youngest samples that's not being properly represented.

Lines 256-257: This looks to be a vague way of saying the boulder surfaces show few signs of erosion, and does not provide any useful information about the condition of

the surfaces. Is it true erosion is not evident? I'm skeptical, as the sentence after this implies there may in fact be considerable erosion. Please provide more detailed descriptions of the quality and appearance of the sampled surfaces in the first paragraph of this section Also, please add some photos of the sampled boulders and surfaces - I would say at least a couple boulder/bedrock photos are required in order to show readers the sample sites.

Line 276 / Figure 3: What is the vertical exaggeration in this figure? Assuming there's none, the Speke moraine would appear to be on a very steep and unstable location right beneath big cliffs that are prone to rockfall. In other words, it looks to be a risky place for exposure dating. This might not be as bad as it looks if the VE (if there is any) was turned down.

Line 301 / Figure 4: See earlier comment. It's very odd to show isotope concentrations rather than apparent ages for the bedrock in this figure. Please show the ages instead.

Lines 324-327: Not sure I agree with this interpretation. Steep ice-contact proximal slopes and more gentle ice-distal slopes are very typical of young / recently abandoned moraines, including those found in locations only minimally or not affected by rockfall. There's no evidence presented here ruling out the possibility of large volumes of debris transported sub- and englacially to the glacier margin as the moraine was being constructed.

Lines 328-329: See earlier comment. How is it known that the sampled boulders were deposited by the glacier, rather than coming from rockfall from the upslope cliffs that came to rest in post-glacial times?

Line 426: Rather than "dominate" consider replacing with "result in negative"

Lines 434-436: You had said earlier that you would not use these two ages in any subsequent interpretations. If that's the intention, this speculation should be omitted here.

Line 445: Replace "fact" with "interpretation"

Lines 511-513: This is a very far-reaching statement to support based on the modest number of new ages presented in this manuscript. The data are especially sparse for the Late Holocene; only 4 ages on one moraine segment are leaned on as being representative of the timing of Late Holocene glaciation in the East African tropics, which is a big extrapolation. And while tempting, it's an even bigger jump to then suggest these ages support a common pan-tropical climate forcing. Apart from the sparse chronology issue, there's also the uncertainty of what specific climate controls are dominating glacier mass balances in various tropical regions on separate continents and over a range of scales from regional to single-valley. The authors favor temperature as the main driver but acknowledge some major untested assumptions, hence a lingering enigma. I encourage the authors to dial it back here, and not go much further than to say their ages hint at similarities in Holocene glacial fluctuations in tropical South America and East Africa, but that a lot more age control (and more modeling, as they suggest) is needed to explore this further.

―――――――――――――――――――

---

## Author Comment (AC1) · 1 Aug 2020

We thank Referee 1 for their thoughtful comments on our manuscript. Although we disagree with certain of the referee's points, we think that their critiques are well-founded and believe that in addressing these concerns our manuscript is much improved.

We have addressed each element of Referee 1's critique in full. In certain cases this entails some revision and modification of the manuscript, but we believe the majority of the Referee's concerns may be addressed through clarifying our intent both here and within the text.

Below we delineate the original referee comments in «brackets» and outline our response below each relevant passage.

« The manuscript "Holocene glaciation in the Rwenzori Mountains, Uganda" by Jackson et al., presents a new set of Holocene cosmogenic dates (n=12) from 2 valleys in Uganda. In my opinion, this study should not be published in the journal Climate of the Past. The primary reasons for this are (1) it is too regional in significance to be appropriate for COP (note the title- does not really reflect an inference about climate); (2) in my opinion it has a very limited ability to provide any concrete climate inferences, not because of the study design, but because of the number of samples, and the quality of the data that can be provided from the limited chronological constraint provided by these dates.; »

We appreciate that Referee 1 feels this manuscript may be a poor fit for Climate of the Past. Indeed, journal 'fit' is never cut-and-dried, and different readers can (and do!) come away with different views on how a contribution fits the overall mission of a publication. We think that these data and the resultant information to be gleaned from them about past glaciation and tropical climate in East Africa are appropriate for Climate of the Past. This is particularly so in consideration of (a) the fact that there exist so few records of past terrestrial temperature from the tropics writ large and (b) our novel coupling of Holocene glacial chronologies with local records of temperature and precipitation. We are also encouraged by the comments from Referees 2 and 3, who support publication of the manuscript in Climate of the Past (pending revision).

The type of direct comparison between past glacial fluctuations and paleoclimate conditions that we present is not possible to undertake elsewhere in the tropics, as similar such terrestrial temperature records (brGDGT geochemical temperature reconstructions) are not available outside of East Africa. Thus, the manuscript presents the first comparison of past tropical glacial fluctuations with nearby temperature records for the Holocene. The interpretations that we are able to make from the Rwenzori data presented here may aid in understanding past glacial chronologies from the wider tropics where such temperature records are not available. In addition, the Rwenzori chronology is particularly useful because it constrains glacial extents for the whole of

the Holocene in multiple catchments, which is not the case for existing East African glacial chronologies. At Kilimanjaro, surface-exposure data do not bear directly on Holocene ice extents (Shanahan and Zreda, 2000). At Mt. Kenya, although there are Holocene-age surface-exposure data for the Teleki valley catchment, these are not replicated in any other catchments. The available surface-exposure data are also scattered (Shanahan and Zreda, 2000), and so are not useful for correlating with other records on millennial or centennial timescales.

We respectfully disagree with the Referee's comment that the dataset presented here is too small to provide any concrete inferences. Although modest in size, the Rwenzori glacial chronology presented here documents glacial extents for the entire Holocene Epoch. We argue that the combined detailed field mapping and lack of samples is itself important and noteworthy, as there are no moraines to be dated in either the Nyamugasani or Bujuku valley between the Early Holocene moraines (∼11 ka) and the dated and/or observed Late Holocene ice extents. This, in addition to the dated perched boulders and other geomorphic evidence discussed, shows rapid Early Holocene glacial retreat and evidence that glaciers remained near or inboard of their Late Holocene/Historical maxima throughout the Holocene Epoch.

Regarding the title of the manuscript, we will revise this to reflect the true scope and novelty of the work. Our suggested/edited title is "Holocene glaciation and climate conditions in East Africa".

« (3) the main conclusions, while theoretically plausible, are not unambiguously supported by the data, and this is not clear from the abstract or conclusions;»

We agree that the language within the manuscript was too definitive in tone and lacked an emphasis on the unknowns yet to be explored. We will modify the language to make explicit the remaining uncertainties and will refocus the latter portion of the discussion to emphasise the Holocene history of glaciation in East Africa. This is also noted below and in our responses to Referees 2 and 3.

« (4) I am concerned that given the reliance on ages that are presented and under review elsewhere (Jackson et al., under review), for the interpretations that are being made in this paper, that really the two papers should have been combined and splitting the manuscripts in two seems unjustified. I explain in more detail below. The paper presents only 12 dates – a small number to constrain any sort of cosmogenic glacial history, particularly since the authors have several other papers published or currently in review from the same sites. Without seeing the other papers, the apparent justification is that this paper is focused on Holocene variability. The authors thus present some older deglacial ages (e.g., 10-12 kyr BP) that appear to be from the other Jackson et al., in press paper, some latest Holocene ages (300-500 yr BP, n=4), in another valley 5 boulders on bedrock – not associated with glacial moraines (11-12 kyr (n=2), ∼4 kyr BP n=3) and samples of bedrock in the uppermost cirque from this valley (5-6 kyr, n=3). »

We agree that citing a paper not yet available to the public (Jackson et al, in review) at the time of submission was not ideal. This paper is now accepted for publication in Quaternary Science Reviews and will be cited as Jackson et al. (2020). We provide a web link to the published journal article here [https://www.sciencedirect.com/science/article/pii/S0277379120304170].

The paper referred to (i.e., Jackson et al., 2020) reports and interprets a Rwenzori glacial chronology for late-glacial time (∼16-11 ka). We intentionally split off the data in the CP manuscript because it deals with a Rwenzori glacial chronology for the Holocene. We felt that the late-glacial and Holocene data required quite different backgrounds and understanding of regional and global climate conditions and dynamics, and the implications of these datasets were different in geographic and climatic scope. As mentioned above, the number of new 10Be ages presented in the CP manuscript, while small, still greatly increases what is known about Rwenzori glaciation during the Holocene and is an important contribution to existing East African records.

« The way I would interpret this data is that it does seem that you had an early Holocene

deglacial retreat at 11-10 kyr BP. And there is evidence from one site that there was a small standstill or readvance during the Little Ice Age (note that this readvance is seen in other African localities, including Mt Kenya and Kilimajaro, I think. The interpretation of the other boulder ages is ambiguous. The upvalley cirque ages of 5 kyr are the same as the valley boulders. So how do we interpret these boulder ages? They are not associated with any geomorphic features, so they may just reflect material deposited during retreat of the ice, and their age may not have any real meaning (ie perhaps they are simply inherited cosmogenic nuclides. Alternatively, the 5kyr ages in the uppermost cirque may suggest that the valleys were basically ice free by 5 kyr. »

We are in agreement with Referee 1 that the data presented suggest rapid deglaciation after ∼11 ka, and apparent readvance or re-nucleation of ice during the Late Holocene (a similar Late Holocene moraine is dated on Mt. Kenya (Shanahan and Zreda, 2000) and discussed in the text). However, we do not consider the older/lower-elevation perched boulder ages (RZ-15-09, RZ-15-10, RZ-15-11) from the Nyamugasani valley to be ambiguous, and suggest that these reflect continued Early Holocene glacial recession (∼11-10 ka). The two other perched boulder ages (RZ-15-07, RZ-15-08) are ambiguous, because (a) the ages are similar to the 'equivalent' exposure ages of the nearby bedrock, and (b) the boulders themselves were covered by ice and/or snow during the early to middle 20th century. We acknowledge the uncertain interpretations of these latter two boulder ages in the text and refrain from basing paleoclimatic interpretations on them. We note that these boulder ages may support a Middle Holocene ice advance to a position near or inboard of the Late Holocene/historical ice extent.

« From this data they make several inferences: 1- glaciers did not readvance beyond their late Holocene maxima during the early or mid Holocene. This is possible, but given that the record is inherently erosive, it is hard to say anything from the absence of evidence. The old ages in the upper cirque bedrock do seem to support this inference (that ice was gone by 5 kyr) but then the authors also say this: Line 355 "Although the timing of ice recession and re-nucleation within the cirque cannot be established

with the data presented here, the bedrock 10Be concentrations suggest that the cirque remained ice-free for a significant portion of the Holocene Epoch." »

We think that this apparent uncertainty from Referee 1 likely reflects our failure to make clear the potential history recorded by bedrock surface-exposure age equivalents, and the way in which we explicitly treat these bedrock data differently than we do exposure ages as measured from perched boulders or moraines. Although we agree it is difficult to prove a negative, we suggest that the lack of evidence for significant glacial expansion outboard of Late Holocene maxima in the Rwenzori is indeed a sufficient, and in fact robust, basis for inferring this glacial history. The absence of additional moraines outboard of the ∼250-450 yr old Speke moraine in the Bujuku valley, in a catchment that provides ample material for moraine formation, is one indication. In the Nyamugasani valley, the perched boulders dated to ∼11-10 ka were, we suggest, deposited by ice retreat/thinning during the Early Holocene. It is unlikely that these boulders were re-covered by a readvance of ice during the Holocene, as a glacier extensive enough to reach this location would have presumably entrained these boulders as it flowed down valley rather than leaving them unaltered. Such downslope transport would have also presumably rotated the boulders such that the exposure ages would not show the similarity they do (∼12-11 ka). Moreover, glacial cover for any appreciable period of time would have reduced the exposure ages of these samples, a possibility which seems unlikely in light of their similarity with samples farther down valley in the catchment.

The bedrock itself yields 10Be concentrations equivalent to ∼5 ka of total exposure. As we note in the text, we hesitate to infer that these measurements reflect a single duration of exposure and so leave open the possibility that ice may have ablated away completely from the cirque and re-nucleated once or many times during the Holocene. Whereas boulders that are entrained during sub- or englacial transport are presumed to undergo erosion sufficient to remove any surface material containing already accumulated 10Be, bedrock surfaces may not always be abraided sufficiently by flowing ice to remove such 'inherited' 10Be (Dunai, 2010).

We make the assumption that the bedrock was sufficiently eroded during the last glacial period to remove any accumulated 10Be in the surface and thus to 'reset' the surface-exposure clock. Although the nature of single-nuclide dating leaves determining the absolute history of this bedrock impossible, we suggest multiple scenarios in the text that may explain the measured 10Be concentrations. These concentrations may, as Referee 1 points out, be the results of a single period of ∼5 ka exposure during the Early, Middle, or Late Holocene. Alternatively, ice may have ablated away and later re-nucleated once or multiple times over the course of the Holocene. We suggest that ice ablated away completely at least once (otherwise the 10Be concentration would be significantly lower) and later renucleated. The fact that ice covered the site in the early and middle 20th century (as observed by Osmaston and Pasteur, 1972, and documented by Mary Meader in 1937) before ablating away indicates this scenario occurred at least once in the Holocene.

« 2- Line 436 "our comparison. . . indicates that ice masses did not respond linearly to temperature." From the really limited data here, I don't see how you can say this. The delay may be simply a lag in the response rather than a nonlinear adjustment. »

We agree that a lagged response may be a contributing factor to the inferred pattern of ice-extent fluctuations and note this in Line 444 of the manuscript.

« 3- Line 552 conclusion: "Based on a comparison of tropical East African glacial fluctuations with regional climate records, we suggest that temperature acted as the primary control on glacial fluctuations throughout the Holocene". I have two concerns about this statement. First is that it is hard to establish the relationship between the glacial chronology and temperature reconstructions given the limited constraints. Essentially what we have is that ice retreat started in the early Holocene and was likely complete by 5 kyr (unless there is cosmogenic inheritance in the upper cirque bedrock). The African lake temperature records do show a signal that is consistent with this but it is not the only explanation. For example, several of the precipitation records also show declining values from 11-12 kyr and could be partially responsible for the glacier retreat. Essentially the problem is that we have very little we can say about what happened during the Holocene from these samples. Furthermore, there is no apparent cooling during the Little Ice Age in the temperature records which could account for the glacial readvance at that time. »

We agree that there are many factors that influence glacial mass balance. As we note in the text, glaciers are sensitive not only to temperature, but to a number of geographic and hydroclimatic variables (Lines 399-404). Relatively low precipitation amounts during the Younger Dryas ($\sim$12.8-11.7 ka) may have contributed to a negative mass balance and glacial retreat, but we note that the onset of the African Humid Period at $\sim$11.6 ka marked a rapid transition to more moist conditions in the region, and all precipitation records we highlight show rapid precipitation rise underway by $\sim$11.4 ka. The Holocene temperature compilation of Ivory et al. (2017) suggests that regional temperatures roughly plateaued between $\sim$11.5 and 9.5 ka, as precipitation first increased and then remained elevated. However there is no evidence that glaciers in either catchment readvanced with the onset of elevated precipitation during the period of sustained, consistent temperatures. In this case, and elsewhere as we highlight in the text, we suggest that although precipitation affected mass balance, at no point in the record were precipitation levels sufficient to overcome the impacts of changing temperature.

Regarding the Little Ice Age, the compilation of regional temperature records produced by Ivory et al. (2017) does show cooling on the order of $\sim$0.5°C between $\sim$700-250 yrs BP. We would argue that this may align with the glacial readvance (and the timing of subsequent recession) recorded in the Rwenzori and at Mt. Kenya $\sim$250 yrs BP.

« Its not clear that the South American comparison is all that convincing of a tropics-wide temperature mechanism. Both Africa and South America chronologies suggest that retreat started early and the most extensive subsequent advance was in the latest Holocene. But South American records show evidence of much greater dynamics (presumably associated with rapid retreat) after 5 kyr, whereas at Ruwenzori, the ice

was already nearly gone by that time. »

We agree that there is much more work to be done assessing the possible centennial-scale synchrony of Holocene glacial fluctuations across the tropics. We will change the tone and text to address the existing uncertainties in the comparison. However, the broad similarities that Referee 1 highlights (i.e., Early Holocene retreat, most extensive subsequent advance during the Late Holocene) are exactly what we focus on in the manuscript. We think these similarities are worth noting.

Although it is beyond the scope of this work, we note that these sorts of broad similarities in regional patterns of deglaciation have been used to compare and contrast glacial records from the Northern and Southern Hemispheres (e.g., Putnam et al., 2012). Glacial chronologies from the European Alps generally indicate rapid Early Holocene retreat and subsequent Middle or Late Holocene re-nucleation/advance. In contrast, glaciers in New Zealand retreated in more stepwise fashion throughout the Holocene. Although suggesting that all Northern Hemisphere glaciers fluctuated synchronously is not possible (nor accurate), the broad similarities are worth noting when glaciers elsewhere in the world display such a markedly different history. To be clear, we agree that our suggestion of a common pan-tropical forcing is likely too speculative and will remove this from the revised text. However, we want to state clearly our interpretations of the existing data from tropical regions, and outline where we may have a different view from Referee 1.

« As a final point, I note that the authors did not appropriately reference the literature in making some of their statements. For example:

Abstract Line27: I think it is incorrect to say that "little is known about the response of tropical glaciers". There is literature from South America for certain, and though I am less familiar, probably in Asia as well. »

We agree that the abstract line highlighted here is too blunt and requires more nuance (we address this also in our responses to Referees 2 and 3). We hope that the edits

and alterations we suggest making to the rest of the manuscript, in addition to our responses from Referees 2 and 3, have reduced Referee 1's concerns in this regard. We include below our response to Referee 2's comment on this passage below:

Response to Referee 2: "We agree that our statement here is blunt and needs nuance. With this statement we are referring only to the relative paucity of data on Holocene tropical glacial fluctuations relative to what is known for higher-latitude glacial fluctuations. Figure 2 in Solomina et al. (2015) provides an illustration of this point. The 'low-latitudes' in this case are 22 data entries on Holocene glacial fluctuations, including one from Papua New Guinea, three from East Africa (one at Kilimanjaro and two from Mt. Kenya), and 18 from South America. Although this is by no means "little" data for the tropics, it is much less than higher-latitude regions. For example, the same data compilation includes eight studies from Spitsbergen and 15 entries from the monsoon-influenced portion of the Himalaya (Solomina et al., 2015). Figure 2 in Solomina et al. (2015) also highlights a fundamental element of many tropical glacial chronologies, namely that many of these entries for tropical glacial fluctuations do not provide information about glacial fluctuations throughout the Holocene, but rather more limited time slices. We think tropical glacial histories are of particular interest due to the relative lack of data from the tropics (and tropical Africa in particular), a point we will clarify in the revised version of the manuscript."

« L411 – Garcin 2007 is not the appropriate reference for the African Humid Period. I suggest referencing some of the early primary literature by Francoise Gasse and other leaders in the field. »

We did not mean to suggest that Garcin et al. (2007) marks the initial recognition of the African Humid Period, nor would we wish to imply such understanding with our readers. The citation included for the African Humid Period (Garcin et al., 2007) is one we chose in order to define the timing and duration of the event based on recent chronological methods and assessments. We propose to alter the relevant text to include earlier work (e.g., Gasse, 2000) to mark the establishment of the African Humid Period in the

literature, while also citing Garcin et al. (2007) as a more recent (temporal) definition of the period that we utilise in our discussion.

References:

Dunai, T.J.: Cosmogenic Nuclides: Principles, concepts and applications in the Earth surface sciences. Cambridge University Press. 2010.

Garcin, Y., Vincens, A., Williamson, D., Buchet, G., and Guiot, J.: Abrupt resumption of the African Monsoon at the Younger Dryas-Holocene climatic transition. Quaternary Science Reviews 26, 690-704. 2007.

Gasse, F. Hydrological changes in the African tropics since the Last Glacial Maximum. Quaternary Science Reviews, 19(1-5), 189-211. 2000.

Ivory, S.J. and Russell, J.: Lowland forest collapse and early human impacts at the end of the African Humid Period at Lake Edward, equatorial East Africa. Quaternary Research 89, 7-20. 2017.

Jackson, M.S., Kelly, M.A., Russell, J.M., Doughty, A.M., Howley, J.A., Cavagnaro, D.B., Zimmerman, S.R.H., and Nakileza, B.: Glacial fluctuations in tropical Africa during the last glacial termination and implications for tropical climate following the Last Glacial Maximum. Quaternary Science Reviews 243, 106455. 2020.

Jomelli, V., Favier, V., Vuille, M., Braucher, R., Martin, L., Blard, P.-H., Colose, C., Brunstein, D., He, F., Khodri, M., BourleİĂs, D.L., Leanni, L., Rinterknecht, V., Grancher, D., Francou, B., Ceballos, J.L., Fonseca, H., Liu, Z., and Otto-Bleisner, L.: A major advance of tropical Andean glaciers during the Antarctic cold reversal. Nature 513, 224-228. 2014.

Putnam, A.E., Schaefer, J.M., Denton, G.H., Barrell, D.J., Finkel, R.C., Andersen, B.G., Schwartz, R., Chinn, T.J. and Doughty, A.M., 2012. Regional climate control of glaciers in New Zealand and Europe during the pre-industrial Holocene. Nature Geoscience, 5(9), 627-630. 2012.

Rodbell, D.T., Seltzer, G.O., Mark, B.G., Smith, J.A., and Abbott, M.B.: Clastic sediment flux to tropical Andean lakes: records of glaciation and soil erosion. Quaternary Science Reviews 27, 1612–1626. 2008.

Shanahan, T. and Zreda, M.: Chronology of Quaternary glaciations in East Africa. Earth and Planetary Science Letters 177, 23-42. 2000.

––––––––––––––––––––

---

## Author Comment (AC2) · 1 Aug 2020

We have addressed each element of Referee 2's comments below. We note where comments from Referee 2 overlap with comments from Referee 1 or 3. We highlight the original referee comments in «brackets» and give our responses below each relevant passage.

« The paper by Jackson et al. provide twelve new 10Be based surface exposure ages from the Rwenzori Mountains of Uganda to place new constraints on late Pleistocene and Holocene glacial changes. Overall, I find the paper easy to read and think the results could be of interest to a broad audience such as the readership of Climate of the Past. However, I think the authors need to address several shortfalls before

the paper is ready for publication. Below I have provided my line number edits and comments. »

We thank Referee 2 for their thoughtful comments and suggestions, which will improve the final manuscript. Below we address each comment individually and include suggested alterations to the text and figures.

« Abstract: - While I agree that understanding tropical glaciation in the past is a worthwhile endeavor, I think the authors have overstated how much we do not know about tropical glaciation over the Holocene. Certainly, there is more than "relatively little" known in my mind. I suggest trying to be more quantitative about what they are referring by "little" so the reader can better understand their argument. »

We agree that our statement here is blunt and needs nuance. With this statement we are referring only to the relative paucity of data on Holocene tropical glacial fluctuations relative to what is known for higher-latitude glacial fluctuations. Figure 2 in Solomina et al. (2015) provides an illustration of this point. The 'low-latitudes' in this case are 22 data entries on Holocene glacial fluctuations, including one from Papua New Guinea, three from East Africa (one at Kilimanjaro and two from Mt. Kenya), and 18 from South America. Although this is by no means "little" data for the tropics, it is much less than higher-latitude regions. For example, the same data compilation includes eight studies from Spitsbergen and 15 entries from the monsoon-influenced Himalaya (Solomina et al., 2015). Figure 2 in Solomina et al. (2015) also highlights a fundamental element of many tropical glacial chronologies, namely that many of these entries for tropical glacial fluctuations do not provide information about glacial fluctuations throughout the Holocene, but rather more limited time slices. We think tropical glacial histories are of particular interest due to the relative lack of data from the tropics (and tropical Africa in particular), a point we will clarify in the revised version of the manuscript.

« Line 171, 199, etc.: The authors cite the work of Jackson et al. in review in multiple places within the paper. I am not sure about CPs policy on citing in review or unpublished peer review work but from a reader's perspective this seems unusual and unhelpful since I cannot reference back to the paper to understand what is being cited. I also think the authors are relying quite a bit on this other data and wonder why the two papers have been separated from one another given these data from the in review paper are somewhat critical to the author's arguments. »

(Here we provide the same response as given to a similar comment by Referee 1): We agree that citing a paper not yet available to the public (Jackson et al, in review) at the time of submission was not ideal. This paper is now accepted for publication in Quaternary Science Reviews and will be cited as Jackson et al. (2020). We provide a web link to the published journal article here [https://www.sciencedirect.com/science/article/pii/S0277379120304170].

The paper referred to (i.e., Jackson et al., 2020) reports and interprets a Rwenzori glacial chronology for late-glacial time (∼16-11 ka). We intentionally split off the data in the CP manuscript because it deals with a Rwenzori glacial chronology for the Holocene. We felt that the late-glacial and Holocene data required quite different backgrounds and understanding of regional and global climate conditions and dynamics, and the implications of these datasets were different in geographic and climatic scope. As mentioned above, the number of new 10Be ages presented in the CP manuscript, while small, still greatly increases what is known about Rwenzori glaciation during the Holocene and is an important contribution to existing East African records.

« Figure 2: It would be useful in Figure 2c if the authors provided some indication of the moraine crest for the reader – perhaps some arrows. »

We will provide additional annotation in this figure and will outline the moraine crest in Figure 2c as well as sample locations in Figure 2b.

« Figure 2: In Figure 2c there appear to be many trees within the photo. I'm confused by this photo given that the authors state on lines 250-252 that vegetation is sparse above 4000 m and that vegetation cover was not used to correct the 10Be dates. Some

explanation seems warranted here since there appears to be clear signs of heavy vegetation cover in this region. Is this the ∼300-400 year Speke moraine, or is it an older moraine? »

This is a photo of the Speke moraine (a view of the right lateral, taken from the left-lateral ridge). The scale in the photo is perhaps misleading, as the 'trees' here are no more than 1-2 m high. We do not consider it necessary to correct for any impact of this vegetation because the shrubs/stalks persist primarily below the ridge crest itself (as visible in Figure 2c). The boulders and cobbles on the ridge crest feature some moss cover. However, based upon the density and thickness of the moss, and using shielding parameters as described by Plug et al. (2007), a persistent moss cover on this moraine ridge for the full duration of exposure would alter the ultimate exposure ages by only ∼2% at most, or ∼10 years. As we think it unlikely that moss was a persistent feature for the full duration of exposure (in terms of ecological succession and moss thickness), we do not correct the 10Be ages for any such vegetation effects. We will clearly explain this reasoning in the text.

« Lines 231: The process blank 9/10Be ratio should be provided as well as the carrier name or stock number. This is important for historical documentation. »

This information (blank 10/9Be ratio) is given in Table 2. We will add the name of the carrier used for each sample group to Table 2.

« Lines 238-240: The authors should say why they choose one over the other and they should say why it does not alter their conclusions. This would help the reader more fully understand their position on the matter. »

We choose to utilise time-invariant "St" scaling because existing high-elevation, low-latitude production rate calibrations are most robust (lowest uncertainty by total scatter) when determined using time-invariant scaling (as evaluated using v3 of the online cosmogenic nuclide exposure-age calculator described by Balco et al. (2008) and subsequently updated). We suggest, however, that although "St" scaling is the most

appropriate choice as based on this metric, the use of an alternative time-variant scaling scheme (such as "LSDn") would not fundamentally alter our conclusions as the difference in 10Be ages calculated with these schemes does not fall beyond the 2-sigma error threshold for either calculation. Moreover, we do not attempt to make any centennial-scale inferences regarding these data. The largest age offsets between scaling methodologies occur in the bedrock 10Be exposure-age equivalents ($\sim$400 years difference when using "St" versus "LSDn" scaling) and we do not make climatic interpretations using these data.

« Line 240-243: While I understand why presenting the ratios might be justified, I think it makes the several parts in the manuscript confusing for the reader doing it this way. Especially, when the authors then use the calibrated ages later on in the study (see the Discussion). I suggest providing the calculated ages and simply explaining why they are likely complex ages related to prior exposure. »

We initially reported these data as ratios rather than as 'exposure ages' in order to prevent readers from perhaps misinterpreting the data when reviewing the figures. We note in the text that it is inadvisable to treat these bedrock ages as 'simple' exposure ages of single duration. However, we understand the need for clarity in the figure, and will change these show the 'exposure age' of these bedrock samples. We will mark these samples in the legend as 'exposure-age equivalent' rather than 'yr BP'.

« Line 248-250: Provide a citation for the daytime temperatures and solar radiation that are being referenced. As-is, I find the snow correction argument weak and it needs more justification which I think a few references could help with. »

Weather station data collected by Lentini et al. (2011) reflects temperature and precipitation variability on Mt. Stanley in the central Rwenzori at an elevation of 4,750 m from the year 2006 to 2009. The Rwenzori glaciers do not experience a seasonal 'winter' snowfall, but instead have a seasonality of precipitation as the ITCZ passes over head twice each year, with greater diurnal temperature changes than any seasonal changes.

Rain falls on the high Rwenzori peaks year round, including during the 'dry' seasons of boreal winter and summer. Although there is some difference in air temperature during the wet and dry seasons, mean daily high air temperatures range between ∼5.5 and 4°C throughout the year. Measured daily low temperatures are between ∼2 and 0°C. In addition, average relative humidity is constant across the seasons (between ∼95-85%). We note that the elevation of the weather station is higher than any of the sample locations we report and describe in our manuscript, and so daytime temperatures at each site should generally reach above 4°C during the day year round. In addition, night time temperatures would rarely be below freezing

« Line 277: In the methods, the authors state that they use the LSDn scaling (Line 239) but throughout the text and starting here they seem to be using the St scaling. This needs to be corrected in the paper and/or table. »

In lines 235-238, we state that we utilise "St" scaling in all figures (and report these in Table 3). We include "LSDn" scaling results in Table 3 for comparison and, while we note that 10Be ages calculated using both scaling methods are similar, we utilise "St" throughout the discussion (see response above). We will clarify this in a revised manuscript.

« Line 288: It would be useful to the reader if the authors provided some photos of the boulder that they are referring or some dimensional information about the boulders. This is important information to convey to the reader since the classification of boulder is large (i.e. > 256 mm). »

We will provide information regarding boulder dimensions and additional field sample photos in a revised manuscript.

« Figure 4: - I recommend providing ages instead of ratios for the three bedrock samples - In the legend and the boxes, it is hard to differentiate between the colors. Either makes these lines thicker and/or make the colors standout more between each other. - The sample ID provided in the boxes do not match the samples numbers from the

tables. I'm therefore not sure what the samples numbers in the boxes represent and would suggest matching them to the tables for reader. - The authors provide data from an in review paper. I find these data quite helpful for their argument but unfortunately because it is in another paper I cannot reasonably evaluate the data. Again, I wonder why the authors have split the data between the papers but think until the in review paper is published it makes these data less convincing to the author's arguments. »

As mentioned above, we initially reported these data as ratios rather than as 'exposure ages' in order to prevent readers from perhaps misinterpreting the data when reviewing the figures. We will change this figure to show the 'exposure age' of these bedrock samples. We will mark these samples in the legend as 'exposure-age equivalent' rather than 'yr BP' and will make the colors/lines more distinct to make the distinction between sample types clearer.

The Sample ID numbers (RZ-XX-XX) are separate from the Map ID number (included in the figures) and refer instead to the original field sample ID and designation as used throughout fieldwork and later laboratory processing. We will make this clearer in the Table captions.

As noted above, the paper we cite as 'in review' is now accepted for publication in Quaternary Science Reviews and will be cited as Jackson et al. (2020). We provide a web link to the published journal article here [https://www.sciencedirect.com/science/article/pii/S0277379120304170].

« Line 313: When the authors say "glacial deposits" do they also mean till? Is it bare bedrock? »

We do not mean till here, necessarily, as the surface of the valley floor has been infilled by wetland (and so is not bare bedrock). Nearer the valley walls (and in some cases transiting the valley floor) colluvial deposits spill into the catchment. In other valleys at similar elevation to the upper Bujuku (3900-4000 m asl), such as in the Nyamugasani valley, although wetland has infilled areas of the valley floor there are moraines clearly

visible above the wetland surfaces. In some cases these moraines dam lakes. We make our observation here about the Bujuku valley based upon the paucity of any such moraines, as well as the absence of glacial till/moraines on the valley walls above the wetlands (in areas not affected by colluvium). We will expand the discussion to make our geomorphic observations more explicit.

« Line 314: It would be useful to see a more zoomed out view of the Bujuku valley so the ~11 ka moraine vs. the Speke moraine can be seen. »

We will update all map figures that include the existing satellite views, to be paired with map-view hill-shaded contour maps of the areas of interest. For the Bujuku valley we will include the ~11.7 ka moraine and the ~11 ka landslide.

« Line 315: Cavagnaro, 2017 is an undergraduate thesis. I'm not sure if it is appropriate to cite work that is not peer reviewed. »

We consider this thesis to be appropriate to cite because it is available online through the Dartmouth College library. However, we will instead cite Jackson et al. (2020), which also includes the landslide data.

« Line 317: While the landslide is not disturbed, is it possible it occurred onto the glacier and then melted out? Without more information about the landslide and how it was dated, it is hard as a reader to evaluate if this is true without having a photo of the landslide or more information beyond the unpublished thesis that I was not able to access. »

We do not think that the landslide occurred atop the glacier, as there is no indication of slope change or disturbance. If the landslide had melted out/settled after initial deposition, we would not expect the sampled landslide boulders to yield similar 10Be ages. Because two samples yield identical 10Be ages, we suggest that any post-depositional movement or rotation of the slide and its sediments is unlikely. We will expand the discussion to make our geomorphic observations more explicit.

« Line 322-325: Here I strongly disagree with the authors. Steep ice-contact slopes and more gentle distal slopes are the norm for moraines and especially true for young moraines that are ice cored and late Holocene in age. Therefore, I don't understand how the authors conclude the moraine was related to rock fall. Is the moraine highly sorted? Is it possible the "moraine" is in fact a protalus rampart (e.g. Ballntyne and Kirkbride, 1986)? »

Here we clarify our description of the Speke moraine. The moraine ridges have sharp crests, with steep ice-contact slopes and more gentle ice-distal slopes. We suggest that the moraine ridges themselves were constructed along the margins of the former Speke glacier at least in part by sediments derived from rockfall onto the glacier surface, although we do not consider this to be the exclusive mechanism of formation. The very steep mountain slopes above the former glacier position would likely have provided material for later deposition via rockfall onto the glacier surface. This is supported in part by the fact that some of the clasts that comprise the Speke moraine are angular rather than molded in form, suggesting they did not undergo extensive subglacial erosion or entrainment. Other clasts, however, show evidence of glacial abrasion, suggesting that this feature is not a protalus rampart. In addition, Ballantyne and Kirkbride (1986) note that the ice-proximal slopes of pro-talus ramparts are generally less steep than ice-distal slopes, which is not the case for the Speke moraine. We will add additional images of this feature to Figure 3 and expand our description to make the context and geometry of the landform clearer.

« Line 326: Based on the imagery I cannot see the fan-like slopes. Is it possible to provide some zoomed in images of what is being referred? »

We will include additional imagery (satellite and field photos) of the Speke moraine in Figure 3.

« Line 327: If the moraine is in fact a debris slide onto the slope, then how can the ages be interpreted as the timing of moraine abandonment? It seems like the boulder

ages should be predating the timing of the glacier retreat. This needs to be explained. »

Here we would suggest that the 'older' Speke moraine ages (∼400-460 yrs BP) may reflect the fact that some moraine material was sourced via rockfall onto the glacier surface and so was minimally eroded. These ages would therefore contain 'inherited' 10Be. In this case the 'younger' ages (∼270-360 yrs BP) would be more representative of the timing of moraine abandonment. However, due to the scatter of ages on this moraine we do not attempt to correlate the timing of moraine abandonment with any discrete climatic forcing, and rather note only that following Early Holocene recession the maximum extent of ice on Mt. Speke occurred during the Late Holocene, within the last ∼400 years.

« Line 339: The ages are now being presented for the bedrock ages. The authors need to decide if they are going to present the ages or the ratios. I suggest the former. »

(Here we provide the same response as given to a similar comment by Referee 2 above): We initially reported these data as ratios rather than as 'exposure ages' in order to prevent readers from perhaps misinterpreting the data when reviewing the figures. We note in the text that it is inadvisable to treat these bedrock ages as 'simple' exposure ages of single duration. However, we understand the need for clarity in the figure, and will change these show the 'exposure age' of these bedrock samples. We will mark these samples in the legend as 'exposure-age equivalent' rather than 'yr BP'.

« Line 340: What evidence is this based on (photos, documentation, etc.)? »

As cited in the text, this is based on direct observations by Whittow (1963) and H. Osmaston (Osmaston and Pasteur, 1972). In addition, Mary Meader (1937) photographed the peak during her aerial survey of the Rwenzori, photos of which are available for viewing online. A link to these photos is included in the works cited [https://collections.lib.uwm.edu/digital/collection/agsafrica/id/358/rec/102].

[Figure]

« Line 357-358: This argument is reasonable but the assumption that LGM ice core erased all prior exposure needs to be restate here for the reader. Also, it would be worth providing some justification about why this assumption is more reasonable than inheritance being pervasive. »

We will re-state this point here for the reader. We will also mention that, although we consider it unlikely, we cannot prove that all bedrock surfaces were sufficiently eroded during the LGM to remove any inherited 10Be. Regardless, we do not rely on the bedrock 'exposure-age equivalents' for any aspect of the discussion beyond the comparison with 20th century ice-cover observations. Thus, any potential inherited 10Be in the bedrock samples/ages does not affect our interpretations and discussion.

« Lines 367: Again, some boulder photos would go a long way. »

We will add additional field and sample photos to the revised manuscript.

« Line 373: The author might comment on why there is no late Holocene moraine at both locations? Is this an elevation or aspect issue? »

This is likely an elevation issue first and foremost. Mt. Speke (4,890 m asl) is higher than Mt. Weisman (4,620 m asl). In addition, the Speke moraine (∼4,050 m asl) represents a much greater catchment area (and sediment delivery area) than does the Thomson cirque (∼4,525). If ice on Mt. Weisman was not effectively erosive, it would take a much longer time to produce a moraine ridge on Mt. Weisman than on Mt. Speke.

« Line 402: Again, it is hard to evaluate new and prior work that is not published yet. »

(Here we provide the same response as given to a similar comment by Referee 2, above): We agree that citing a paper not yet available to the public (Jackson et al, in review) at the time of submission was not ideal. This paper is now accepted for publication in Quaternary Science Reviews (QSR) and will be cited as Jackson et al. (2020). We provide a web link to the published journal article here

[https://www.sciencedirect.com/science/article/pii/S0277379120304170].

« Line 437: It is not clear what the authors mean. Presumably the glaciers respond to temperature in the same way – more temperature means more ablation and vice versa. I think other factors come into play here (hypsometry, winter precipitation, energy balance) and therefore it isn't that they respond non-linearly it is that other factors are important and that temperature is not the only driver of ice mass position. »

We agree that temperature is not the sole control on glacial mass balance in the Rwenzori or elsewhere, and did not intend for that to be implied. Precipitation, hypsometry, humidity, aspect, all have a role to play in mass balance change, and disentangling each variable from the others is difficult even with high-resolution models. However, we do want to clarify what we mean in this portion of the discussion.

In using the term 'non-linear' to describe the inferred pattern of glacial fluctuations in the Rwenzori, we wanted to highlight the apparent threshold response of ice in the Bujuku and Nyamugasani valleys to Early Holocene warming. Although precipitation remained elevated from ~11.7-5 ka, this elevated precipitation regime was not sufficient to maintain positive mass balance - or indeed to allow glaciers to persist. In addition, the apparent expansion of glaciers in the Late Holocene coincides with more recent apparent cooling with persistent dry conditions. Although it may be inappropriate to draw direct correlations between recent decadal changes and millennial-scale changes during the Holocene (as boundary conditions such as greenhouse gas levels and insolation were different), these records suggest that temperature acted (and may yet act) as the primary control on mass balance, and that any future increase in regional precipitation would be insufficient to induce glacial regrowth or readvance in light of still warming temperatures.

« Line 538-540: If the glaciers are similar tropic wide, then I would expect other climate archives to also reflect this. What about the PAGES 2k reconstructions or the Marcott et al. 2013 stacks? Do they show a similar pattern as suggested by the glacial data?

These needs further justification. »

In the revised text we will step back from our suggestion that glaciers in the African and South American tropics experienced synchronous fluctuations during the Holocene. We acknowledge that the data are not yet robust enough in both regions to make this statement. However, Referee 2's comment does bring up an interesting point regarding the existence and availability of tropical temperature records beyond tropical Africa.

The Marcott et al. (2013) data stacks show relatively little change in Holocene temperatures in the low latitudes. This reconstruction suggests that Early Holocene temperatures were only ∼0.4°C cooler relative to the Holocene average, and that a Middle Holocene climatic optimum centred at ∼5 ka was only ∼0.15° C warmer than average. In contrast, the regional African temperature stack of Ivory et al. (2017) indicates temperature changes of ∼2.5°C between ∼11 and 5 ka. The overall pattern of temperature changes reported by Marcott et al. (2013) and Ivory et al. (2017) is similar, but the magnitude and variability of the former is much less than that of the latter. This may be due to the predominance of marine proxies used by Marcott et al. (2013). In contrast, the record of Ivory et al. (2017) is based entirely on terrestrial proxies and may indicate the influence of lapse rate changes on terrestrial (and higher elevation) temperatures over time relative to marine temperature changes. In any case, the differences between these proxy types are an important avenue for future study. The similarity of sign, if not magnitude, between the Marcott et al. (2013) stack and the Ivory et al. (2017) compilation may point toward a broader similarity in wider tropical temperatures, but more work is needed to assess this possibility – particularly in the terrestrial tropics.

References:

Ballantyne, C.K. and Kirkbride, M.P. The characteristics and significance of some Lateglacial protalus ramparts in upland Britain. Earth Surface Processes and Landforms, 11(6), pp.659-671. 1986.

[Figure]

Balco, G., Stone, J.O., Lifton, N.A. and Dunai, T.J.: A complete and easily accessible means of calculating surface exposure ages or erosion rates from 10Be and 26Al measurements. Quaternary Geochronology 3, 174-195. 2008.

Ivory, S.J. and Russell, J.: Lowland forest collapse and early human impacts at the end of the African Humid Period at Lake Edward, equatorial East Africa. Quaternary Research 89, 7-20. 2017.

Jackson, M.S., Kelly, M.A., Russell, J.M., Doughty, A.M., Howley, J.A., Cavagnaro, D.B., Zimmerman, S.R.H., and Nakileza, B.: Glacial fluctuations in tropical Africa during the last glacial termination and implications for tropical climate following the Last Glacial Maximum. Quaternary Science Reviews 243, 106455. 2020.

Jomelli, V., Favier, V., Vuille, M., Braucher, R., Martin, L., Blard, P.-H., Colose, C., Brunstein, D., He, F., Khodri, M., BourleÌĂs, D.L., Leanni, L., Rinterknecht, V., Grancher, D., Francou, B., Ceballos, J.L., Fonseca, H., Liu, Z., and Otto-Bleisner, L.: A major advance of tropical Andean glaciers during the Antarctic cold reversal. Nature 513, 224-228. 2014.

Lentini, G., Cristofanelli, P., Duchi, R., Marinoni, A., Verza, G., Vuillermoz, E.L., Toffolon, R. and Bonasoni, P., 2011. Mount Rwenzori (4750 m asl, Uganda): meteorological characterization and Air-Mass transport analysis. Geografia Fisica e Dinamica Quaternaria, 34(2), 183-193. 2011.

Marcott, S.A., Shakun, J.D., Clark, P.U. and Mix, A.C.. A reconstruction of regional and global temperature for the past 11,300 years. Science, 339(6124), 1198-1201. 2013.

Osmaston, H.A. and Pasteur, D.: Guide to the Ruwenzori. Alden Press, Oxford, pp. 200. 1972. Plug, L.J., Gosse, J.C., McIntosh, J.J. and Bigley, R. Attenuation of cosmic ray flux in temperate forest. Journal of Geophysical Research: Earth Surface, 112(F2). 2007.

Solomina, O.N., Bradley, R.S., Hodgson, D.A., Ivy-Ochs, S., Jomelli, V., Mackinstosh,

A.N., Nesje, A., Owen, L.A., Wanner, H., Wiles, G.C., and Young, N.E.: Holocene glacier fluctuations. Quaternary Science Reviews 111, 9-34. 2015.

Whittow, J.B., Shepherd, A., and Goldthorpe, J.E.: Observations on the glaciers of the Ruwenzori. Journal of Glaciology 4, 581-616. 1963.
* * *

---

## Author Comment (AC3) · 1 Aug 2020

We have addressed each element of Referee 3's comments below. Where comments from Referee 3 overlap with comments from Referee 1 or 2 we make note. We highlight the original referee comments in «brackets» and outline our response below each relevant passage.

«This short manuscript is clearly written and well-organized, and reports a small but interesting new set of twelve Be-10 ages from glacial moraines, boulders, and bedrock surfaces in the Rwenzori Mountains. The authors present some reasonable interpretations of the Holocene glacial history at their field sites based on their data and field observations. These results are then compared with regional climate proxies and other

glacial records in East Africa, and also to tropical glacier records in South America. Overall, I think these newly reported findings from the Rwenzori are valuable and add important knowledge to the glacial and climate history of tropical East Africa. However, the number of new exposure ages is quite modest, and as such, it is difficult to extrapolate from these to make strong arguments about commonalities with Holocene glacier records in South America and pan-tropical climate forcings. I support the publication of these results after appropriate revisions, but I urge the authors to be more cautious and realistic about the limitations of inferring global-scale correlations and climate forcing mechanisms from a small data set.

More specific comments and critiques are listed below. I hope the authors find these constructive, and I encourage them to address and resolve these in order to improve their manuscript. »

We thank Referee 3 for their thoughtful comments and agree that, while our assessment of the Rwenzori data in the context of regional climate records is robust, some inferences were overly speculative in terms of the broader, pan-tropical synchrony of Holocene glacial fluctuations. With this in mind (and with similar comments from Referees 1 and 2, see replies posted), we propose to refocus the final portion of our discussion to emphasise tropical East African glacial fluctuations during the Holocene and comparisons of tropical East African glacial chronologies with local climate (i.e., terrestrial temperature and precipitation) records. We will illustrate the utility of these glacial records in studying tropical East African paleoclimate conditions.

Below we outline our responses to each of Referee 3's comments, and our proposed alterations/improvements to the manuscript in each case where applicable.

« Lines 27-28: That's probably not a fair statement these days, at least outside Africa, as there have been a number of studies and reviews of tropical glaciation in recent years. »

(Here we provide the same response as given to a similar comment by Referees 1 and

2): We agree that our statement here is blunt and needs nuance. With this statement we are referring only to the relative paucity of data on Holocene tropical glacial fluctuations relative to what is known for higher-latitude glacial fluctuations. Figure 2 in Solomina et al. (2015) provides an illustration of this point. The 'low-latitudes' in this case are 22 data entries on Holocene glacial fluctuations, including one from Papua New Guinea, three from East Africa (one at Kilimanjaro and two from Mt. Kenya), and 18 from South America. Although this is by no means "little" data for the tropics, it is much less than higher-latitude regions. For example, the same data compilation includes eight studies from Spitsbergen and 15 entries from the monsoon-influenced Himalaya (Solomina et al., 2015). Figure 2 in Solomina et al. (2015) also highlights a fundamental element of many tropical glacial chronologies, namely that many of these entries for tropical glacial fluctuations do not provide information about glacial fluctuations throughout the Holocene, but rather more limited time slices. We think tropical glacial histories are of particular interest due to the relative lack of data from the tropics (and tropical Africa in particular), a point we will clarify in the revised version of the manuscript.

« Line 68: List in chronological order. It's also curious that the Late Holocene is not regarded here as a time period of interest - especially in light of statements about this work's relevance to modern/future climate change. »

We will change this sentence to introduce the Early and Middle Holocene periods in descending chronologic order. By stating here that the warm Early and Middle Holocene are periods of particular interest, we do not mean to suggest that the Late Holocene and modern period is not of interest. However, we think that investigating the past response of glaciers to warmer conditions is especially important within the larger context of modern warming and glacial sensitivity to climate change. We will clarify this in the text.

« Line 82 / Figure 1: The satellite image in panel b is not an acceptable substitute for a proper glacial-geomorphic map. The moraines and overall topography are very

hard to see. I suggest replacing with a DEM or contour map if available, overlain by a more detailed map indicating glacial features and other relevant landforms in these valleys. Without a well-labeled glacial-geomorphic map, the text descriptions of these field areas are very hard to follow. »

We will replace Panel B in Figure 2 with a hill-shaded contour map with geomorphic features highlighted. We will also add similar map-view, hill-shaded contour maps to Figures 3 and 4 to highlight valley sample locations.

« Line 93: That is very inclusive. What kind of crystalline rock, exactly? »

The rocks of the Rwenzori include Precambrian metamorphic rocks, predominantly gneisses and schists, as well as amphibolites (McConnell, 1959; Ring, 2008; Bauer, 2010). The Nyamugasani valley is composed of quartz-rich gneiss, whereas the Bujuku valley is composed on its south side by amphibolite and on its north side (including Mt. Speke) by quartz-bearing schist and gneiss. We will add this information to the updated manuscript.

« Line 111: This is the first of many citations to an in-review manuscript that is not currently accessible to reviewers. Because many interpretations here are reliant on context and support from the results in the unavailable in-review manuscript, it is not really possible to properly assess this new manuscript. In fact, the frequent references to the in-press manuscript and the importance of those findings to the interpretation of the new ages reported here raises the question of why these data were not all reported together in a single paper. »

(Here we provide the same response as given to a similar comment by Referees 1 and 2): We agree that citing a paper not yet available to the public (Jackson et al, in review) at the time of submission was not ideal. This paper is now accepted for publication in Quaternary Science Reviews (QSR) and will be cited as Jackson et al. (2020). We provide a web link to the published journal article here [https://www.sciencedirect.com/science/article/pii/S0277379120304170].

The paper referred to (i.e., Jackson et al., 2020) reports and interprets a Rwenzori glacial chronology for late-glacial time (∼16-11 ka). We intentionally split off the data in the CP manuscript because it deals with a Rwenzori glacial chronology for the Holocene. We felt that the late-glacial and Holocene data required quite different backgrounds and understanding of regional and global climate conditions and dynamics, and the implications of these datasets were different in geographic and climatic scopes. As mentioned above, the number of new 10Be ages presented in the CP manuscript, while small, still greatly increases what is known about Rwenzori glaciation during the Holocene and is an important contribution to existing East African records.

« Line 123: I assume these calendar ages are recalibrated from the original radiocarbon data using updated calibration curves, but the details of the calendar age estimation need to be explained here. »

We recalculated the original published 14C ages using the IntCal13 radiocarbon curve (Reimer et al., 2013) and the Calib 7.1 calculator (Stuiver et al., 2020). Ages are reported as the midpoint age, with 2-sigma uncertainty. We will make note of the radiocarbon age calibration and presentation and expand upon this in the Methods section.

« Line 190: How is the landslide dated? How reliable is that age? »

The landslide is dated using10Be dating of boulder surfaces on the landslide. Three 10Be ages from boulders near the toe of the landslide yielded ages between ∼11 and 12 ka, with two samples returning near-identical ages of ∼11 ka. These data were first reported in the Dartmouth College Senior Honors Thesis of Cavagnaro (2017). The data are also in the Jackson et al. (2020) available here: [https://www.sciencedirect.com/science/article/pii/S0277379120304170].

« Line 229 / Table 1: Density and erosion columns can be eliminated since the values are uniform for all samples. Instead, just note these values in a footnote. Also, how close are the three boulders (RZ-12-22, 24, 25) with indistinguishable latitude-longitude

coordinates? A field photo would help show the field relations. »

We include density and erosion here so that readers can easily 'copy and paste' these data into the online calculator used to calculate these reported exposure ages (v2 and v3 of the online calculator as described by Balco et al. (2008) and subsequently updated). We will eliminate the columns here as suggested and include them in the 'copy-paste' version of the table in an online supplement. We will include additional field photos of this moraine in Figure 3 to help aid in reader understanding and interpretation of the landform and the samples collected/discussed.

« Lines 241-244: It's an odd choice to show Be-10 concentrations instead of apparent ages for the bedrock surfaces on Figure 4, even if there's a suspicion of complex exposure scenarios. This forces readers to find the ages in Table 3 (where they are reported) to gain some sense of the exposure durations and how they fit in with the other ages on the map. Also, if isotope inheritance is the concern, then that same issue could also potentially apply to the boulder surfaces - as acknowledged in the discussion. »

(Here we provide the same response as given to a similar comment by Referee 2): We initially reported these data as ratios rather than as 'exposure ages' in order to prevent readers from perhaps misinterpreting the data when reviewing the figures. We note in the text that it is inadvisable to treat these bedrock ages as 'simple' exposure ages of single duration. However, we understand the need for clarity in the figure, and will change these show the 'exposure age' of these bedrock samples. We will mark these samples in the legend as 'exposure-age equivalent' rather than 'yr BP'.

« Line 246 / Table 2: Why are the isotope ratios in a different table than the concentrations (and the ages, for that matter)? I suggest some consolidation of the three tables, ideally into one table if possible. The first three columns are identical in all of them, other columns can be eliminated (as noted earlier), and it's inconvenient to have to retrieve data from individual samples spread across three different tables. Also, given that the sample ratios are just over one order of magnitude above the blanks, it is im-

portant to consider how well these blank values are known. If they are all prepared from the same spike, it appears they vary quite a bit - and are therefore known with far less certainty than implied by the analytical uncertainties on individual blanks. This is a potentially important source of uncertainty for the youngest samples that's not being properly represented. »

We included data in tables as would be required for simple 'copy-paste' use and comparison when calculating or recalculating the 10Be ages. We can consolidate the data within these tables for ease of reference within the text, and will provide data in a 'copy-paste' format in a supplement online. The ratios of the process blanks vary. The 12 samples reported here were processed in four separate batches over time, and thus reflect four separate process blanks (as listed in Table 2). We will reformat the tables for ease of use and interpretation.

« Lines 256-257: This looks to be a vague way of saying the boulder surfaces show few signs of erosion, and does not provide any useful information about the condition of the surfaces. Is it true erosion is not evident? I'm skeptical, as the sentence after this implies there may in fact be considerable erosion. Please provide more detailed descriptions of the quality and appearance of the sampled surfaces in the first paragraph of this section Also, please add some photos of the sampled boulders and surfaces - I would say at least a couple boulder/bedrock photos are required in order to show readers the sample sites. »

We will add additional field and sample photos to the manuscript in order to give a better indication of the conditions and contexts of sample surfaces. We will also expand on our description of sample surfaces and apparent erosion (or lack thereof).

« Line 276 / Figure 3: What is the vertical exaggeration in this figure? Assuming there's none, the Speke moraine would appear to be on a very steep and unstable location right beneath big cliffs that are prone to rockfall. In other words, it looks to be a risky place for exposure dating. This might not be as bad as it looks if the VE (if there is any)

was turned down. »

We will add additional field and satellite photos of this site to Figure 3 in order to provide greater geomorphic context for the reader. We note that we were careful to select clasts for surface-exposure dating that occurred on the crest of the Speke moraine, were stable/rooted in till, and showed no evidence of post-depositions alteration/movement or potential deposition by direct rockfall.

« Line 301 / Figure 4: See earlier comment. It's very odd to show isotope concentrations rather than apparent ages for the bedrock in this figure. Please show the ages instead. »

(Here we provide the same response as given to a similar comment by Referee 3 above): We initially reported these data as ratios rather than as 'exposure ages' in order to prevent readers from perhaps misinterpreting the data when reviewing the figures. We note in the text that it is inadvisable to treat these bedrock ages as 'simple' exposure ages of single duration. However, we understand the need for clarity in the figure, and will change these show the 'exposure age' of these bedrock samples. We will mark these samples in the legend as 'exposure-age equivalent' rather than 'yr BP'.

« Lines 324-327: Not sure I agree with this interpretation. Steep ice-contact proximal slopes and more gentle ice-distal slopes are very typical of young / recently abandoned moraines, including those found in locations only minimally or not affected by rockfall. There's no evidence presented here ruling out the possibility of large volumes of debris transported sub- and englacially to the glacier margin as the moraine was being constructed. »

We will clarify that our suggestion of (some) rockfall contribution to the moraine is based on the presence of highly angular, apparently unweathered clasts that likely did not undergo any subglacial abraision/erosion. However, as there were some clasts that showed glacial abrasion, we infer that rockfall was not the sole source of material for the moraine.

« Lines 328-329: See earlier comment. How is it known that the sampled boulders were deposited by the glacier, rather than coming from rockfall from the upslope cliffs that came to rest in post-glacial times? »

(See prior comment/response above regarding Lines 324-327).

« Line 426: Rather than "dominate" consider replacing with "result in negative" »

We will alter the language here as suggested.

« Lines 434-436: You had said earlier that you would not use these two ages in any subsequent interpretations. If that's the intention, this speculation should be omitted here. »

We note in the text that we do not use these two boulder ages for subsequent interpretations. However, in light of the apparent similarity between the calculated exposure ages for these samples and the exposure-age equivalents for the bedrock samples upslope, we think it is necessary to remark on the similarity and the potential for these ages to reflect 'inherited' 10Be while acknowledging that a counter-argument may be made that these sample reflect a Middle Holocene readvance of ice on Mt. Weisman (absent inherited 10Be). We do not make any attempt to correlate these samples with broader paleoclimate records, or with other glacial fluctuations elsewhere in the tropics. We will make our (lack of) reliance on the ages clearer in the discussion and in this passage.

« Line 445: Replace "fact" with "interpretation" »

We will alter the language in this sentence to make clear that it is based on our interpretation/inference rather than a known 'fact'.

« Lines 511-513: This is a very far-reaching statement to support based on the modest number of new ages presented in this manuscript. The data are especially sparse for the Late Holocene; only 4 ages on one moraine segment are leaned on as being representative of the timing of Late Holocene glaciation in the East African tropics, which is

a big extrapolation. And while tempting, it's an even bigger jump to then suggest these ages support a common pan-tropical climate forcing. Apart from the sparse chronology issue, there's also the uncertainty of what specific climate controls are dominating glacier mass balances in various tropical regions on separate continents and over a range of scales from regional to single-valley. The authors favor temperature as the main driver but acknowledge some major untested assumptions, hence a lingering enigma. I encourage the authors to dial it back here, and not go much further than to say their ages hint at similarities in Holocene glacial fluctuations in tropical South America and East Africa, but that a lot more age control (and more modeling, as they suggest) is needed to explore this further. »

We agree that there is certainly much more work to be done assessing the possible centennial-scale synchrony of Holocene glacial fluctuations across the tropics. We will change the tone and text to address the existing uncertainties in the comparison.

To address Referee 3's comment regarding the Late Holocene, we will make the remaining uncertainties in the timing and magnitude of regional glacial fluctuations clearer, although we note that the Late Holocene age of the Lewis Glacier moraine ($\sim$ 210 yrs BP) dated by Shanahan and Zreda (2000) is similar to the age of the Speke moraine we report from the Rwenzori.

References:

Balco, G., Stone, J.O., Lifton, N.A. and Dunai, T.J.: A complete and easily accessible means of calculating surface exposure ages or erosion rates from 10Be and 26Al measurements. Quaternary Geochronology 3, 174-195. 2008.

Bauer, F.U., Glasmacher, U.A., Ring, U., Schumann, A. and Nagudi, B. Thermal and exhumation history of the central Rwenzori Mountains, Western rift of the east African rift system, Uganda. International Journal of Earth Sciences, 99(7), 1575-1597. 2010.

Jackson, M.S., Kelly, M.A., Russell, J.M., Doughty, A.M., Howley, J.A., Cavagnaro, D.B.,

Zimmerman, S.R.H., and Nakileza, B.: Glacial fluctuations in tropical Africa during the last glacial termination and implications for tropical climate following the Last Glacial Maximum. Quaternary Science Reviews 243, 106455. 2020.

McConnell, R.B.: Outline of the geology of the Ruwenzori Mountains, a preliminary account of the results of the British Ruwenzori expedition, 1951-1952. Overseas Geology and Mineral Resources 7, 245-268. 1959.

Putnam, A.E., Schaefer, J.M., Denton, G.H., Barrell, D.J., Finkel, R.C., Andersen, B.G., Schwartz, R., Chinn, T.J. and Doughty, A.M., 2012. Regional climate control of glaciers in New Zealand and Europe during the pre-industrial Holocene. Nature Geoscience, 5(9), 627-630. 2012.

Ring, U. Extreme uplift of the Rwenzori Mountains in the East African Rift, Uganda: Structural framework and possible role of glaciations. Tectonics, 27(4). 2008.

Solomina, O.N., Bradley, R.S., Hodgson, D.A., Ivy-Ochs, S., Jomelli, V., Mackinstosh, A.N., Nesje, A., Owen, L.A., Wanner, H., Wiles, G.C., and Young, N.E.: Holocene glacier fluctuations. Quaternary Science Reviews 111, 9-34. 2015.
* * *

---

## Author Comment (AC4) · 1 Aug 2020

First and foremost we thank you for your thoughtful comments on our manuscript. Your comments echo many of those made by Referee 1 (the response to whom is posted for review), and certain from Referees 2 and 3 (see responses). We've endeavoured to address each of your questions and comments here. We include your original comments «in brackets» and outline our responses below, clarifying our intent or outlining the planned alterations we will make to the manuscript.

« Following previous publications (Kelly et al., 2014; Jackson et al., 2019; Jackson et al., in review), this manuscript is the fourth one presenting glacial chronological data from more or less similar sites in the Rwenzori Mountains. Splitting a glacial chronology

into several papers might be reasonable to have more space for discussing individual aspects/events in detail, but the added value of this manuscript remains unclear after first reading. Most of the exposure ages from the Nyamugasani Valley (Fig. 4) that determine the glacier extent at ∼12-11 ka (before the onset of deglaciation) stem apparently from the other manuscript in review (Jackson et al., in review). From my understanding, the "only" new finding based on the additional ages from the upper part of the valley is that the "Thomson cirque" (and maybe Mount Weisman too?) was probably ice-free by ∼5 ka or at least for a longer period during the Holocene. Since the reader has no insight into the other manuscript in review, it would be important to elucidate the novelty or new aspect of the contribution presented here. »

The new data we present come from two Rwenzori valleys, the Nyamugasani and the Bujuku valleys. These data provide the first direct constraints on past ice extents in both valleys throughout the Holocene Epoch. They show a pattern of similar glacial fluctuations in both valleys – specifically rapid Early Holocene recession of ice to within the Late Holocene maximum glacial extents. In addition to this new information regarding Rwenzori glaciation, we make use of local records of paleotemperature and precipitation in order to evaluate the response of Rwenzori glaciers to Holocene climatic changes. Such an analysis cannot be conducted elsewhere in the tropics, as terrestrial temperature records such as those from East Africa (e.g., brGDGT temperature reconstructions from lake sediments) do not yet exist for other regions. We also synthesise existing literature on East African glaciation during the Holocene in order to provide a review of past glaciation in the region. Such a review is a useful, novel addition to the current literature, as much existing work on Holocene glaciation in the African tropics was produced before the advent of surface-exposure dating (see references in manuscript). Moreover, existing, global syntheses of Holocene glaciation do not include many of these African records (see Solomina et al., 2015). Although we make these points in the manuscript, we will adjust the language to make these elements of the manuscript and its novelty clearer.

Regarding the paper 'in review', we agree that citing a paper not yet available to the public (Jackson et al, in review) at the time of submission was not ideal. This paper is now accepted for publication in Quaternary Science Reviews and will be cited as Jackson et al. (2020). We provide a web link to the published journal article here [https://www.sciencedirect.com/science/article/pii/S0277379120304170].

The paper referred to (i.e., Jackson et al., 2020) reports and interprets a Rwenzori glacial chronology for late-glacial time (∼16-11 ka). We intentionally split off the data in the CP manuscript because it deals with a Rwenzori glacial chronology for the Holocene. We felt that the late-glacial and Holocene data required quite different backgrounds and understanding of regional and global climate conditions and dynamics, and the implications of these datasets were different in geographic and climatic scope. As mentioned above, the number of new 10Be ages presented in the CP manuscript, while small, still greatly increases what is known about Rwenzori glaciation during the Holocene and is an important contribution to existing East African records.

« In the abstract, the authors propose that "understanding how tropical glaciers responded to past periods of warming is crucial for predicting and adapting to future climate change [...]" (lines 26-27). They state further in the introduction that "tropical glaciers are a primary source of freshwater and are a fundamental component of regional economies [...]" (lines 52-53) and that "determining when and how glaciers in the African tropics fluctuated during past warm periods provides crucial information for assessing whether, or how long, tropical glaciers may persist under future warming scenarios" (lines 70-72). Although the ongoing glacial melting in the tropics and most of the mountains worldwide is of great concern, the contribution of the meltwater from the relatively small glaciers in equatorial Eastern Africa to the alpine runoff is negligible (e.g. Kaser et al., 2004; Taylor et al., 2009) and thus do not seem to play a major role for the regional economy and freshwater supply. »

In the abstract and introduction we outline the many ways that tropical glacial systems play vital roles in the welfare of communities, be it through tourism, as critical parts of

alpine ecosystems, as freshwater reservoirs, or as sources of hydropower. No single tropical glacial system may fall into each of these categories, yet we suggest that understanding a system in one valley or region can have implications for understanding tropical glacial systems elsewhere. We also think it useful to highlight the utility and urgency in understanding tropical glacial systems more broadly.

With this in mind, we did not suggest that Rwenzori glaciers (or African glaciers in general) are vital sources of freshwater to local communities, as they are not. However, Rwenzori glaciers underpin regional tourism and are a primary draw for trekkers and tourists visiting the region, as are the remaining glaciers on Mt. Kenya and Kilimanjaro. As such they are a fundamental aspect of local economies. They are also a critical part of the African alpine ecosystem, and the loss of these glaciers has major potential impacts on surrounding habitats (Oyana and Nakileza, 2016).

« Moreover, the authors do not explain how limited information on past glacial fluctuations could help to better project the future evolution of tropical glaciers in response to global climate change. Reconstructed glacier extents and established glacial chronologies provide without doubt important information on past glacier dynamics, but I think the palaeoclimatic, -environmental, and -glacial data are too uncertain to draw meaningful conclusions about "[. . .] the sensitivity of tropical glaciers to future climate change" (lines 545-546). I would even argue the other way round that modern observations and investigations regarding the climate sensitivity of tropical glaciers in Africa are inevitable for a reliable interpretation of past glacier fluctuations in the region (e.g. Mölg et al., 2003; Mölg et al., 2004; Mölg et al., 2008; Mölg et al., 2009; Nicholson et al., 2013). It is a bit surprising that the authors do not pick up the topic again in the discussion and do not emphasize the claimed relevance of their findings for the future evolution of tropical glaciers. I would therefore recommend that the authors rather stress the palaeoclimatic and -environmental relevance of their study in the abstract and introduction. »

We agree that assessing the modern sensitivity of glaciers in the tropics - and elsewhere - to climatic variables is crucial for understanding these systems over time, both in the past and future. However, we would argue that it is not possible to extrapolate fully the dynamics of past glaciation based on modern observations which span only a few decades, particularly when past climate conditions were so different from modern in terms of both global (e.g., greenhouse gas concentrations) and regional (humidity/precipitation, insolation, seasonality, etc.) conditions. While an understanding of modern change is crucial, it can only be made complete, we argue, when contextualised. We will alter the manuscript to make the connection of our study to modern change more explicit. Specifically, we will emphasise our interpretation that temperature played a dominant role in Holocene ice extents despite increases in precipitation. This has implications for future projections of tropical glacial recession in light of precipitation changes.

« The two main conclusions of the manuscript are that (1) Holocene glacier fluctuations were similar across the tropics and based on the consideration of regional climate records that (2) "[. . .] temperature acted as the primary control on glacial fluctuations throughout the Holocene" (lines 553-554). I do not agree with these statements for the following reasons:

1. The 10Be exposure ages from the Holocene moraine stages in the Rwenzori Mountains (Nyamugasani Valley), on Mount Kenya (Teleki Valley), and on Kilimanjaro (Kibo Peak) originate more or less from one valley/locality (Shanahan and Zreda, 2000). Whether the respective ages are representative for the entire mountain range can neither be confirmed nor refuted. I think without further evidence it remains hypothetical whether glaciers in tropical Eastern Africa responded synchronously to Holocene climate changes or not. »

We agree that more work is needed to assess the potential synchrony (or asynchrony) of Holocene glacial fluctuations in East Africa, particularly on centennial timescales. This highlights a vital avenue for future research across the region, one we will emphasise in the manuscript. However, in the case of the Rwenzori, the data are from two

independent glacial catchments and yield similar results, suggesting that the overall pattern we identify is likely representative of the Rwenzori as a whole.

We also suggest that the available evidence from the region, while limited, supports the hypothesis that glacial fluctuations were broadly similar across the region during the Holocene Epoch. The data presented and summarised in the manuscript also represent the sum of the last ∼40 years of work on East African glacial extents during the Holocene. To be sure, there can be no definitive statement at present, but the available evidence is compatible with a broadly coherent regional pattern of glacial fluctuations on millennial timescales. We will make this nuance clearer in the manuscript.

« 2. The Early and Middle Holocene moraine stages dated in the Rwenzori Mountains (∼11.7 ka), on Mount Kenya (∼10.2 and ∼8.6 ka), and on Kilimanjaro (∼13.8 ka) show by no means a similar pattern, apart from a general warming trend after the last glacial period. The differences could be explained by dating uncertainties, but also by climatic variations. How do you interpret the differences? »

A direct comparison of the Holocene moraine data from Kilimanjaro and Mt. Kenya with the Rwenzori chronology requires a full re-calculation of the original 36Cl surface-exposure ages (Shanahan and Zreda, 2000) to incorporate updated production rate and production-rate scaling methodologies. However, this is not possible because the sample data required to perform such a recalculation are not included in the original publication (Shanahan and Zreda, 2000). Although the existing 36Cl chronologies from Kilimanjaro and Mt. Kenya are not necessarily in conflict with the glacial chronology from the Rwenzori, we consider it inappropriate to make specific, centennial-scale interpretations based upon the 36Cl ages. In addition, the scatter in ages from single landforms is too great to permit centennial or millennial-scale correlations with discrete climate events. However we do not want to ignore these data outright and consider them worthy of mention in our broader discussion. We suggest that, similar to the Rwenzori chronology, the data broadly indicate Late Glacial and Early Holocene glacial recession on Kilimanjaro and Mt. Kenya.

As regards differences between the Kilimanjaro and Mt. Kenya chronologies, at Mt. Kenya the ages mentioned (∼10.2 and 8.6 ka) come from the Teleki Valley, and older, pre-Holocene ages come from the separate Gorges Valley (Shanahan and Zreda, 2000). Shanahan and Zreda (2000) did not date additional Holocene or Late Glacial landforms farther down valley in the Teleki Valley, nor did they date landforms farther up valley in the Gorges Valley. At Kilimanjaro, the age mentioned (∼13.8 ka) is pre-Holocene, and so does not bear directly on Holocene ice fluctuations. Although there are likely some differences in glacial chronologies between these two mountains for a number of regions (e.g., valley hypsometries, alpine versus plateau glaciation, etc.) whether there is truly a mismatch may only be determined by deliberate sampling and dating of the full suite of landforms in and between valley systems.

« 3. In view of a lacking robust Holocene glacial chronology for Eastern Africa, the dynamic Holocene glacier fluctuations in South America, and the non-consideration of other tropical glacial chronologies, claiming "[similar] Holocene glacial fluctuations across the tropics" (lines 38-39) seems rather speculative than evidence-based. Moreover, this assumption underrates the complex regional response of alpine glacier to climatic changes (e.g. variations in temperature, precipitation, cloudiness, insolation, and moisture) in general. Mountain height, terrain, hypsometry, debris cover, glacier size, and many other geological, geomorphological, glaciological, and climatic parameters control the magnitude and rate of glacier fluctuations, as the regional variations in the response of alpine glaciers to recent global warming underline (e.g. Zemp et al., 2019). »

We absolutely agree that temperature alone cannot account for the fluctuations of tropical glaciers in the Rwenzori or elsewhere, either now or in the past. It is not our intent to argue that temperature is the sole control on glacial mass balance. Indeed, all glaciers are sensitive to a litany of factors, including hypsometry, insolation, and precipitation. We concur that our language within the text was too definitive in this regard, and will alter our text to make the nuance of our argument more clear. However,

we suggest that there are marked similarities between glacial fluctuations across the humid tropics, specifically a marked Early Holocene retreat after ∼11 ka, and apparent re-advance or renucleation of ice during the Late Holocene. Our aim in this text is not to argue for a true synchrony in tropical glacial fluctuations over the course of the Holocene, as indeed there is much work to be done to determine the millennial or centennial-scale fluctuations of glaciers at all sites. Instead we highlight the broad pattern of glacial fluctuations in the tropics that appears more similar than dissimilar over millennial timescales, which may be an intriguing focus for future research.

In our response to Referee 1, we note: "Although it is beyond the scope of this work, we note that these sorts of broad similarities in regional patterns of deglaciation have been used to compare and contrast glacial records from the Northern and Southern Hemispheres (e.g., Putnam et al., 2012). Glacial chronologies from the European Alps generally indicate rapid early Holocene retreat and subsequent Middle or Late Holocene re-nucleation/advance. In contrast, glaciers in New Zealand retreated in more stepwise fashion throughout the Holocene. Although suggesting that all Northern Hemisphere glaciers fluctuated synchronously is not possible (nor accurate), broad similarities are worth noting when glaciers elsewhere in the world display such a markedly different history."

« 4. A key assumption for the author's hypothesis that " [. . .] tropical glaciers responded to a common, pan-tropical forcing mechanism during the Holocene" (lines 534-535) is that tropical glaciers are highly sensitive to changes in temperature (lines 399-404; see also Jackson et al., 2019). As a reference for the Rwenzori Mountains, the authors quote a controversial study (Taylor et al., 2006a; Taylor et al., 2006b) which claims that rising temperatures are the dominant factor for recent glacier melting in the Rwenzori Mountains. However, the detailed comment on this study by Mölg et al. (2006), which elaborates the importance of other climate variables for the energy and mass balance of tropical glaciers, is neglected in the discussion. Multiple studies from Kilimanjaro and the Rwenzori Mountains stress that climate variables related to

air moisture (e.g. specific humidity affecting sublimation, cloudiness affecting incoming solar radiation, precipitation affecting glacier surface albedo and mass gain) play an important role in the present surface energy balance of tropical glaciers in Eastern Africa, especially at high elevations above the 0C isotherm (Mölg et al., 2003; Mölg et al., 2004; Mölg et al., 2006; Mölg et al., 2008; Mölg et al., 2009; Nicholson et al., 2013). Since the sensitivity of tropical glaciers in Eastern Africa to different climate variables is an ongoing and very important debate that is crucial for the hypothesis and conclusions of the presented manuscript, the controversial arguments should find more attention in the discussion. In view of the modern observations, I doubt that past glacial fluctuations in Eastern Africa can and should be explained by temperature variations alone. »

In our response to Referee 2, we wrote: "Relatively low precipitation amounts during the Younger Dryas (∼12.8-11.7 ka) may have contributed to a negative mass balance and glacial retreat, but we note that the onset of the African Humid Period at ∼11.6 ka marked a rapid transition to more moist conditions in the region, and all precipitation records we highlight show rapid precipitation rise underway by ∼11.4 ka. The Holocene temperature compilation of Ivory et al. (2017) suggests that regional temperatures roughly plateaued between ∼11.5 and 9.5 ka, as precipitation first increased and then remained elevated. However there is no evidence that glaciers in either catchment readvanced in time with the onset of elevated precipitation during the period of sustained, consistent temperatures. In this case, and elsewhere as we highlight in the text, we suggest that although precipitation affected mass balance, at no point in the record were precipitation levels sufficient to overcome the impacts of changing temperature."

We agree that temperature alone cannot account for the fluctuations of tropical glaciers in the Rwenzori and elsewhere. It is not our intent to argue that temperature is the sole control on glacial mass balance (see comment above). We suggest that, over millennial timescales, Rwenzori glacial fluctuations reflect a pattern of growth and decay that

does not align with reconstructed regional precipitation during the Holocene, nor would it appear that glaciers readvanced during periods of elevated or rising precipitation.

The studies by Mölg et al. are concerned with recent, decadal-scale changes in glacial extents. Although these are important works which we will be sure to include in our revised manuscript, our chronology cannot speak to decadal-scale changes over the Holocene - just as these other studies mentioned cannot address millennial-scale change.

Taylor et al. (2006) suggests that Rwenzori glacial melt is dominated by changes in atmospheric temperature. Although disputed by other studies (Molg et al., 2003, 2006), the work by Taylor et al. (2006) does not stand alone in its suggestion that recent glacial melt in the Rwenzori is a temperature-dominated signal. For example, Russell et al. (2009) use sedimentary analysis of Rwenzori lakes to infer the onset of recent (near-historical) Rwenzori deglaciation was underway before start of regional drying ~1880 AD. This analysis suggests that glaciers retreated in response to a forcing beyond aridity. To be sure, regional drying after ~1880 AD would have impacted glacial mass balance and likely encouraged further recession. However, whether it is appropriate to make direct comparisons between modern decadal-scale climatology and the conditions of the Early or Middle Holocene, when global boundary conditions were markedly different, is another matter for discussion.

« Specific comments: » « Fig. 1b: Would it be possible to add geographic coordinates to the map of the central Rwenzori massif? »

Yes, we will add coordinates to Figure 1b within the revised figure.

« Fig. 1b: For me as a reader who is not familiar with the region, it is difficult to interpret the terrain on the Worldview-1 satellite image. Replacing the image by a combination of DEM and hillshade (including the shapes of the lakes) might be an alternative. »

We will update Figure 1b within the text in order to make the terrain and geomorphic

context clearer. We plan to insert a hill-shaded contour map of the area of interest in place of the satellite image.

« Fig. 2: Could you include at least one photo of a sampled boulder and bedrock surface so that the reader gets a better impression of the investigated landforms? »

Yes, we will provide photos of the cirque bedrock and of perched erratic boulders within the upper Nyamugasani valley, and will include these in the revised manuscript.

« Lines 135-138 and 425-427: The authors rely solely on radiocarbon ages from lake Garba Guracha in the Bale Mountains to discuss the potential timing of deglaciation in the southern Ethiopian Highlands, although direct 36Cl surface exposure ages of 21 moraine boulders from two valleys in the Bale Mountains are published (see Fig. 1 and S6-8 in Ossendorf et al., 2019). The inner-most moraines in the two valleys show that deglaciation in the Bale Mountains began after ∼15-14 ka and suggest (not necessarily imply) that the southern Ethiopian Highlands were ice-free before the Pleistocene-Holocene transition. »

We regret not highlighting the important work of Ossendörf et al. (2019) within our overview of East African glaciation in our original draft. In emphasising the record from Lake Garba Guracha we intended only to utilise a record that explicitly referenced the Holocene period within its analysis. The work of Ossendörf et al. (2019) detailed glacial fluctuations throughout the last glacial period in the Bale Mountains, but did not include data on potential Holocene fluctuations. We will update the manuscript to include this contextual data from the Ethiopian Highlands.

« Lines 254-256.: Did you conduct a simple sensitivity analysis (assuming e.g. two or three plausible erosion rates) to assess the age uncertainty related to erosion? »

We did not include an erosion-sensitivity analysis, as prior work in the Rwenzori suggests that raised quartz veins and bulk boulder surfaces yield statistically indistinguishable ages (Jackson et al., 2019). However, we will include such an analysis and discussion in a revised manuscript and describe the results of such a sensitivity test here:

For each sample, we calculated 10Be ages as determined with the following rates of erosion using version 3 of the online calculator as described by Balco et al. (2008 and subsequently updated): 0.00, 0.0001, and 0.0003 cm/yr (i.e., between 0 and 3 cm erosion per 10,000 years).

Results of this analysis indicate that for the Speke moraine samples, no erosion rate was capable of altering the calculated exposure age from the zero-erosion scenario. In the case of the Thomson cirque bedrock, the maximum erosion rate scenario utilised yielded exposure ages only ∼1% older than the zero-erosion scenario (∼80 years).

For the Nyamugasani perched boulder transect, the maximum erosion scenario altered ages by ∼2-3% (∼70-300 years), with 'older' calculated exposure ages affected more by potential erosion effects. The more moderate scenario (0.0001 cm/10 kyr) yielded age offsets of ∼110 years, less than 1% of the total exposure age. We note that in each case, the impacts of erosion would not alter our interpretations within the manuscript. Due to the lack of information on and uncertainty surrounding erosion rates in the Rwenzori, we refrain from explicitly including these calculations within the manuscript, but would gladly make note of these values within the Methods section in order to make explicit the negligible impact of potential erosion on the Rwenzori chronology.

« Table 1: Content-wise, the columns with the 10Be concentrations in Table 1 would fit better in Table 2. Information of sample lithology could be included in Table 1 if available. »

We present our reported data in three Tables with a view toward their easy re-use. Table 1 includes all of the information required to immediately 'cut and paste' into the online calculator used for age calculation (v2 and v3 of the online calculator as described by Balco et al. (2008) and subsequently updated), whereas Table 2 delineates the laboratory processing data required for determining the 10Be concentration used for calculation. We are hesitant to blend these tables, but propose to do so for inclusion

[Figure]

in the manuscript text and to provide a secondary, 'cut and paste' version of these same data for download online. See also our replies to Referee 2 regarding table formatting.

« Table 3: Could you outline how you define the "internal" (probably analytical) and "external" error? »

Internal error is the analytical uncertainty attached to a given measurement. External uncertainty includes the uncertainties associated with the chosen nuclide production rate and scaling scheme used to calculate the resultant exposure ages, as well as the uncertainty associated with sample-specific variables such as topographic shielding. We will make this distinction explicit within the table caption.

« Fig. 4: I understand why you report 10Be concentrations instead of exposure ages for the bedrock samples (RZ-15-01, RZ-15-02, RZ-15-03), but they are difficult to inter­pret and compare with the other results. Therefore, I would recommend to report the exposure ages (instead of concentrations) in the map and note in the legend that they indicate the net duration of bedrock exposure, as you outlined in the text. »

(See our response below to Referees 2 and 3 regarding a similar comment): "We initially reported these data as ratios rather than as 'exposure ages' in order to prevent readers from perhaps misinterpreting the data when reviewing the figures. We note in the text that it is inadvisable to treat these bedrock ages as 'simple' exposure ages of single duration. However, we understand the need for clarity in the figure, and will change these to show the 'exposure age' of these bedrock samples. We will mark these samples in the legend as 'exposure-age equivalent' rather than 'yr BP'."

« Fig. 4 and Table 3: Considering the general uncertainties associated with surface exposure dating (analytical errors, unknown erosion rates, etc.), I don't see justification to report ages in a way (e.g. 11,020 $\pm$ 280 years) that implies the method is precise enough to date events to a specific decade. I would recommend to round the ages and report them in kiloyears (e.g. 11.0 $\pm$ 0.3 ka). »

[Figure]

We agree that uncertainties inherent to surface-exposure dating do not enable decadal certainty in age calculation for reporting, and throughout the Discussion we round these ages to kilo years. However, we believe it is important to report individual ages and associated error, without rounding, within the Results, figures, and data tables in order to highlight the analytical agreement (or disagreement) between discrete samples or landforms. Reporting ages in this way also ensures transparency for those readers who may wish to re-calculate exposure ages or to compare ages as calculated using different production rate scaling schemes. Rounded 'kilo year' ages may obscure the differences between methodologies, and can make later re-interpretation of published results more difficult.

« Section 6.3: What is the rationale to explicitly discuss the glacial fluctuations in tropical South America here, although no new or recalculated ages are presented? The aim/motivation for the exclusive comparison between the Holocene ages from the Rwenzori Mountains and Andes is not clear from the abstract and introduction. Glacial chronological data also exist from other locations across the tropics. »

There are indeed glacial chronological data from locations elsewhere in the tropics, however we choose to focus on data from low-latitude South America for the following reasons:

1) The aim of this comparison is to assess, if possible, the potential similarities/differences in glacial fluctuations from different regions of the low latitudes. Any identifiable difference or similarity has direct bearing on the potential mechanisms that controlled glacial mass balance in the tropics, in East Africa and elsewhere, and thus in reconstructing wider tropical paleoclimate. As noted in response to referee comments, we agree that our original language in this section of the discussion was too certain in tone, and suggest re-focusing this portion of the manuscript to emphasise what is known (and unknown) regarding the African chronologies.

2) The climatic setting of the Rwenzori, in the humid 'inner' tropics (Kaser and Osmaston, 2002) is more similar to the setting of the low-latitude Andes than to other tropical regions which experience a more monsoonal or arid climate (such as the Indian subcontinent or the subtropical Andes). The marked lack of thermal seasonality - and relatively muted seasonality in precipitation - in the humid tropics is a key factor in considering the controls on glacial mass balance (e.g., Sagredo et al., 2014; Rupper and Roe, 2008).

3) We likewise chose to focus our analysis on glaciation in the tropical Andes as this is the region where the majority of prior work on Holocene glaciation has been conducted, and so is the region which provides the most information on past mass balance change as a whole. There is little chronologic control on glacial deposits in Papua New Guinea, and so while we can certainly add mention glaciers at these sites in our revised manuscript, they are of limited use for drawing wider comparisons.

References:

Balco, G., Stone, J.O., Lifton, N.A. and Dunai, T.J.: A complete and easily accessible means of calculating surface exposure ages or erosion rates from 10Be and 26Al measurements. Quaternary Geochronology 3, 174-195. 2008.

Ivory, S.J. and Russell, J.: Lowland forest collapse and early human impacts at the end of the African Humid Period at Lake Edward, equatorial East Africa. Quaternary Research 89, 7-20. 2017.

Jackson, M.S., Kelly, M.A., Russell, J.M., Doughty, A.M., Howley, J.A., Chipman, J.W., Cavagnaro, D., Nakileza, B. and Zimmerman, S.R.: High-latitude warming initiated the onset of the last deglaciation in the tropics. Science Advances, 5(12), eaaw2610. 2019.

Jackson, M.S., Kelly, M.A., Russell, J.M., Doughty, A.M., Howley, J.A., Cavagnaro, D.B., Zimmerman, S.R.H., and Nakileza, B.: Glacial fluctuations in tropical Africa during the last glacial termination and implications for tropical climate following the Last Glacial

Maximum. Quaternary Science Reviews 243, 106455. 2020.

Kaser, G. and Osmatson, H. Tropical Glaciers. Cambridge, Cambridge University Press, 207 pp. 2002. Mölg, T., Hardy, D.R. and Kaser, G.: Solar‐radiation‐maintained glacier recession on Kilimanjaro drawn from combined ice‐radiation geometry modeling. Journal of Geophysical Research: Atmospheres 108 (D23). 2003.

Mölg, T., Rott, H., Kaser, G., Fischer, A. and Cullen, N.J. Comment on "Recent glacial recession in the Rwenzori Mountains of East Africa due to rising air temperature"by Richard G. Taylor, Lucinda Mileham, Callist Tindimugaya, Abushen Majugu, Andrew Muwanga, and Bob Nakileza. Geophysical Research Letters, 33(20), L20404. 2006.

Oyana, T.J. and Nakileza, B.R. Assessing adaptability and response of vegetation to glacier recession in the afro-alpine moorland terrestrial ecosystem of Rwenzori Mountains. Journal of Mountain Science, 13(9), 1584-1597. 2016.

Ossendorf, G., Groos, A.R., Bromm, T., Tekelemariam, M.G., Glaser, B., Lesur, J., Schmidt, J., Akçar, N., Bekele, T., Beldados, A. and Demissew, S.. Middle Stone Age foragers resided in high elevations of the glaciated Bale Mountains, Ethiopia. Science, 365(6453), 583-587. 2019.

Putnam, A.E., Schaefer, J.M., Denton, G.H., Barrell, D.J., Finkel, R.C., Andersen, B.G., Schwartz, R., Chinn, T.J. and Doughty, A.M., 2012. Regional climate control of glaciers in New Zealand and Europe during the pre-industrial Holocene. Nature Geoscience, 5(9), 627-630. 2012.

Rupper, S. and Roe, G.. Glacier changes and regional climate: A mass and energy balance approach. Journal of Climate, 21(20), pp.5384-5401. 2008.

Russell, J.M., Eggermont, H.E., Taylor, R., and Verschuren, D.: Paleolimnological records of recent glacier recession in the Rwenzori Mountains, Uganda-D.R. Congo. Journal of Paleolimnology 41, 253-271. 2009.

Sagredo, E.A., Rupper, S., and Lowell, T.V.: Sensitivities of the equilibrium line altitude

to temperature and precipitation changes along the Andes. Quaternary Research 81, 355–366. 2014.

Shanahan, T. and Zreda, M.: Chronology of Quaternary glaciations in East Africa. Earth and Planetary Science Letters 177, 23-42. 2000.

Solomina, O.N., Bradley, R.S., Hodgson, D.A., Ivy-Ochs, S., Jomelli, V., Mackinstosh, A.N., Nesje, A., Owen, L.A., Wanner, H., Wiles, G.C., and Young, N.E.: Holocene glacier fluctuations. Quaternary Science Reviews 111, 9-34. 2015.

Taylor, R.G., Mileham, L., Tindimugaya, C., Majugu, A., Muwanga, A., and Nakileza, B.: Recent glacial recession in the Ruwenzori Mountains of East Africa due to rising air temperature. Geophysical Research Letters 33, L10402. 2006.